# Three-dimensional mapping of the altermagnetic spin splitting in CrSb

Guowei Yang [1,10], Zhanghuan Li [2,10], Sai Yang [3], Jiyuan Li [3], Hao Zheng [1], Weifan Zhu [1], Ze Pan [1], Yifu Xu [1], Saizheng Cao [1], Wenxuan Zhao [4], Anupam Jana [5,6], Jiawen Zhang [1], Mao Ye [7], Yu Song [1], Lun-Hui Hu [1], Lexian Yang [4], Jun Fujii [5], Ivana Vobornik [5], Ming Shi [1], Huiqiu Yuan [1,8,9], Yongjun Zhang [3] ✉, Yuanfeng Xu [1] ✉ & Yang Liu [1,8] ✉

Altermagnetism, a kind of collinear magnetism that is characterized by a momentum-dependent band and spin splitting without net magnetization, has recently attracted considerable interest. Finding altermagnetic materials with large splitting near the Fermi level necessarily requires three-dimensional $k$-space mapping. While this is crucial for spintronic applications and emergent phenomena, it remains challenging. Here, using synchrotron-based angle-resolved photoemission spectroscopy (ARPES), spin-resolved ARPES and model calculations, we uncover a large altermagnetic splitting, up to ~1.0 eV, near the Fermi level in CrSb. We verify its bulk-type $g$-wave altermagnetism through systematic three-dimensional $k$-space mapping, which unambiguously reveals the altermagnetic symmetry and associated nodal planes. Spin-resolved ARPES measurements further verify the spin polarizations of the split bands near Fermi level. Tight-binding model analysis indicates that the large altermagnetic splitting arises from strong third-nearest-neighbor hopping mediated by Sb ions. The large band/spin splitting near Fermi level in metallic CrSb, together with its high $T_N$ (up to 705 K) and simple spin configuration, paves the way for exploring emergent phenomena and spintronic applications based on altermagnets.

A new type of collinear magnetism, dubbed altermagnetism, has been proposed theoretically[1–11] and verified experimentally in MnTe[12–17]. There have also been experiments supporting altermagnetism in RuO$_2$[18–23], although the nature of magnetism in RuO$_2$ is still debated[24,25]. Altermagnet is often defined in the absence of spin-orbit coupling (SOC) and can be classified by the spin space group, different from ferromagnet and

conventional antiferromagnet. Although altermagnets exhibit zero net magnetization in real space, they host spin-split electronic bands in reciprocal space. While the zero bulk magnetization allows for ultrafast manipulation with low stray field, akin to an antiferromagnet, the time-reversal symmetry breaking and momentum-dependent spin polarization in altermagnets can give rise to emergent properties useful for device

[1]Center for Correlated Matter and School of Physics, Zhejiang University, Hangzhou, China. [2]Beijing National Laboratory for Condensed Matter Physics, Institute of Physics, Chinese Academy of Sciences, Beijing, China. [3]Hubei Key Laboratory of Photoelectric Materials and Devices, School of Materials Science and Engineering, Hubei Normal University, Huangshi, China. [4]State Key Laboratory of Low Dimensional Quantum Physics, Department of Physics, Tsinghua University, Beijing, China. [5]CNR-IOM, TASC Laboratory, Area Science Park-Basovizza, Trieste, Italy. [6]International Center for Theoretical Physics (ICTP), Trieste, Italy. [7]Shanghai Synchrotron Radiation Facility, Shanghai Advanced Research Institute, Chinese Academy of Sciences, Shanghai, China. [8]Collaborative Innovation Center of Advanced Microstructures, Nanjing University, Nanjing, China. [9]State Key Laboratory of Silicon and Advanced Semiconductor Materials, Zhejiang University, Hangzhou, China. [10]These authors contributed equally: Guowei Yang, Zhanghuan Li. ✉e-mail: yjzhang@hbnu.edu.cn; y.xu@zju.edu.cn; yangliuphys@zju.edu.cn

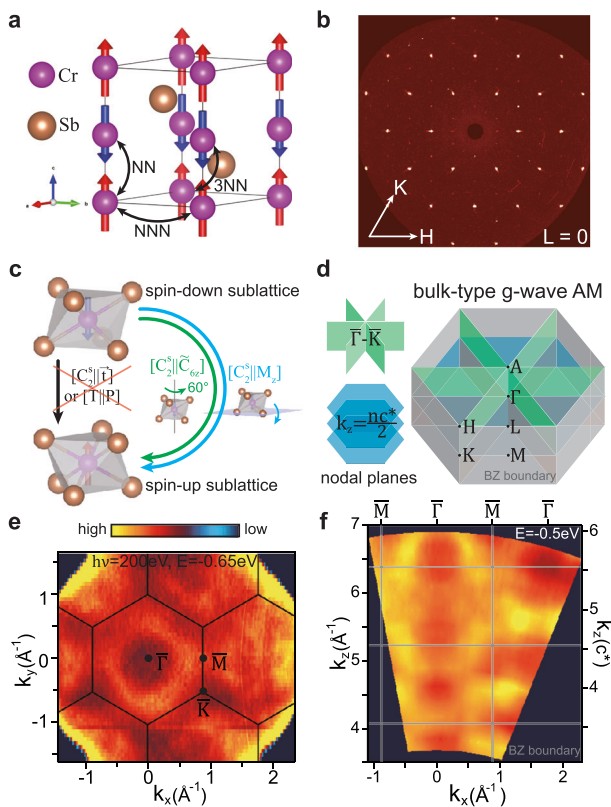

**Fig. 1 | Spin/lattice structure and characterization of CrSb. a** Spin and lattice structure of CrSb. The nearest neighbor (NN), next nearest neighbor (NNN), and third nearest neighbor (3NN) Cr-Cr bonds are indicated by the black arrow lines. **b** H-K map (L = 0) of a (001)-oriented crystal from XRD. **c** Symmetry operations connecting the opposite-spin sublattices. **d** Left: the nodal planes corresponding to spin-group symmetries in (**c**). The $k_z = nc^*/2$ nodal planes, where $c^* = 2\pi/c$ and $n$ is an integer, are protected by $[C_2^s\|M_z]$ combining with translation symmetry. Right: The three-dimensional Brillouin zone (BZ) with high symmetry points and nodal planes labeled. **e** The in-plane $k_x$-$k_y$ map at $E = -0.65$ eV, taken with 200 eV photons (corresponding to $k_z \sim 0.5c^*$). **f** The $k_x$-$k_z$ map at $E = -0.5$ eV along $\bar{\Gamma} - \bar{M}$ ($k_x$). The black hexagons in (**e**) and grey rectangles in (**f**) label the BZ boundaries. The surface BZ for (001)-oriented CrSb, which is the projection of three-dimensional BZ onto the (001) surface, is labelled in (**e**) with the high-symmetry points $\bar{\Gamma}$, $\bar{M}$ and $\bar{K}$. The color bar for image plots in (**e**, **f**) is indicated.

applications, such as anomalous/nonlinear Hall effects[17–19,26–30], giant/tunneling magnetoresistance[31,32], spin currents, spin-torque effects[20–22,33–36], spin caloritronics (Nernst effect)[37–39]. Recent theoretical studies further show that the inclusion of SOC can induce topological states[40,41]. In addition, unconventional superconductivity based on altermagnetic materials has been theoretically proposed[42–47], making them a fertile playground for exploring superconducting states. The momentum-dependent spin splitting has also been found recently in the non-coplanar system MnTe[48], which significantly expands the scope of altermagnetic materials.

The characteristic momentum-dependent spin splitting in an altermagnet results from its special spin-group symmetry[10,49], i.e., the opposite-spin sublattices are connected by a real-space rotation or mirror operation, instead of a translation or inversion. Depending on the dimensionality and lattice/spin symmetry, altermagnets can be classified into planar-type or bulk-type with $d$-, $g$- or $i$-wave symmetry[9], corresponding to two, four or six spin-degenerate nodal planes crossing the $\Gamma$ point, respectively. Since the nodal planes often coincide with high-symmetry momentum planes, it is important to investigate the altermagnetic splitting in three-dimensional momentum space. While altermagnetic splittings have been observed

experimentally in MnTe, how the altermagnetic splitting evolves in three-dimensional momentum space remains unexplored. Most importantly, for enhanced physical properties and real applications, it is critical to find strong altermagnets with large spin splittings near the Fermi level ($E_F$) and with magnetic ordering temperature well above room temperature.

Here in this paper, we present direct spectroscopic evidence of the largest altermagnetic band splitting reported so far, up to ~1.0 eV near $E_F$ in CrSb, from synchrotron-based angle-resolved photoemission spectroscopy (ARPES) and spin-resolved ARPES measurements, combined with ab initio calculations and tight-binding (TB) model analysis. CrSb is traditionally known for its itinerant antiferromagnetism with a high Néel temperature up to 705 K[50]. Although altermagnetic band splitting was recently reported in (100)-oriented CrSb films by soft X-ray ARPES[51], high-resolution three-dimensional $k$-space mapping of the altermagnetic splitting with spin resolution, which is key to confirm (and understand) the altermagnetism proposed in CrSb[9], is still lacking. Our high-resolution ARPES data from (001)-oriented single crystals unambiguously reveal the characteristic $k_z$ and in-plane momentum dependence of the altermagnetic splitting in CrSb. Moreover, our spin-resolved ARPES measurements provide direct evidence of the expected spin polarization near $E_F$. The experimental results are in good agreement with the ab initio calculations from density functional theory (DFT). We further demonstrate through TB model analysis that the large altermagnetic splitting stems from the strong third-nearest-neighbor hoppings of Cr 3$d$ orbitals mediated by the Sb 5$p$ orbitals, which break both the space-time reversal symmetry and the translational spin-rotation symmetry. Such insight can be important for designing and tuning altermagnetic materials with desirable properties.

## Results
### Sample characterization and lattice/spin structure
CrSb crystallizes in the hexagonal NiAs-type structure. As shown in Fig. 1a, each Cr atom is coordinated to the top and bottom Sb triangles with AB stacking, forming a tilted octahedron. The sharp X-ray diffraction (XRD) pattern from a (001)-oriented crystal, shown in Fig. 1b, confirms the high crystal quality with the extracted lattice constants of $a = b = 4.12$ Å, $c = 5.44$ Å, close to the previously reported value of $a = 4.12$ Å and $c = 5.47$ Å[50]. The magnetic structure of CrSb, confirmed by a number of studies based on neutron scattering[50,52–54], belongs to the magnetic space group $P6_3/m'm'c$, where the magnetic moments of Cr (aligned along the [001] axis) are parallel within the hexagonal Cr layer and antiparallel between two neighboring layers. Ignoring the SOC, which is weak for Cr 3$d$ electrons and plays a minor role in the low-energy band structure (see below), the altermagnetism in CrSb can be well defined and characterized by non-relativistic spin-group symmetries. As defined in[9,10,55–59], a spin space group operation is a combination of spin rotation operation and real-space transformation, namely $[R_i^s\|R_j]$, where $R_i^s$ acts only in spin space (containing only identity transformation $E$ and spin-flip operation $C_2^s$ for collinear magnetic systems), while $R_j$ operates in real space. The two opposite-spin sublattices in altermagnets should be connected by a (screw) $n$-fold rotation ($C_n$, $n$ depending on lattice/spin symmetries) or (glide) mirror ($M$) symmetry in real space, rather than by translation ($\vec{t}$) or inversion ($P$) symmetry. As illustrated in Fig. 1c, the two Cr sublattices with opposite spins in CrSb are related by a spin group symmetry $[C_2^s\|M_z]$ or $[C_2^s\|\tilde{C}_{6z}]$, where $\tilde{C}_{6z}$ is a six-fold rotation along $z$ axis combined with a translation $\vec{t} = (0, 0, \frac{1}{2})c$. Both the spin group symmetries $[C_2^s\|\vec{t}]$ and $[T\|P]$ (space-time reversal) are broken in CrSb, resulting in momentum-dependent spin splitting in the electronic band structure. As shown in Fig. 1d, on the mirror invariant planes, i.e., the horizontal planes at $k_z = 0$ and $0.5c^*$ ($c^* = 2\pi/c$), and three equivalent vertical planes containing $\bar{K} - \bar{\Gamma} - \bar{K}$ related to $C_{3z}$, the spin splitting is zero. Therefore, CrSb can be classified as a three-dimensional (or bulk-type) $g$-wave altermagnet with four nodal planes crossing the $\Gamma$ point[9].

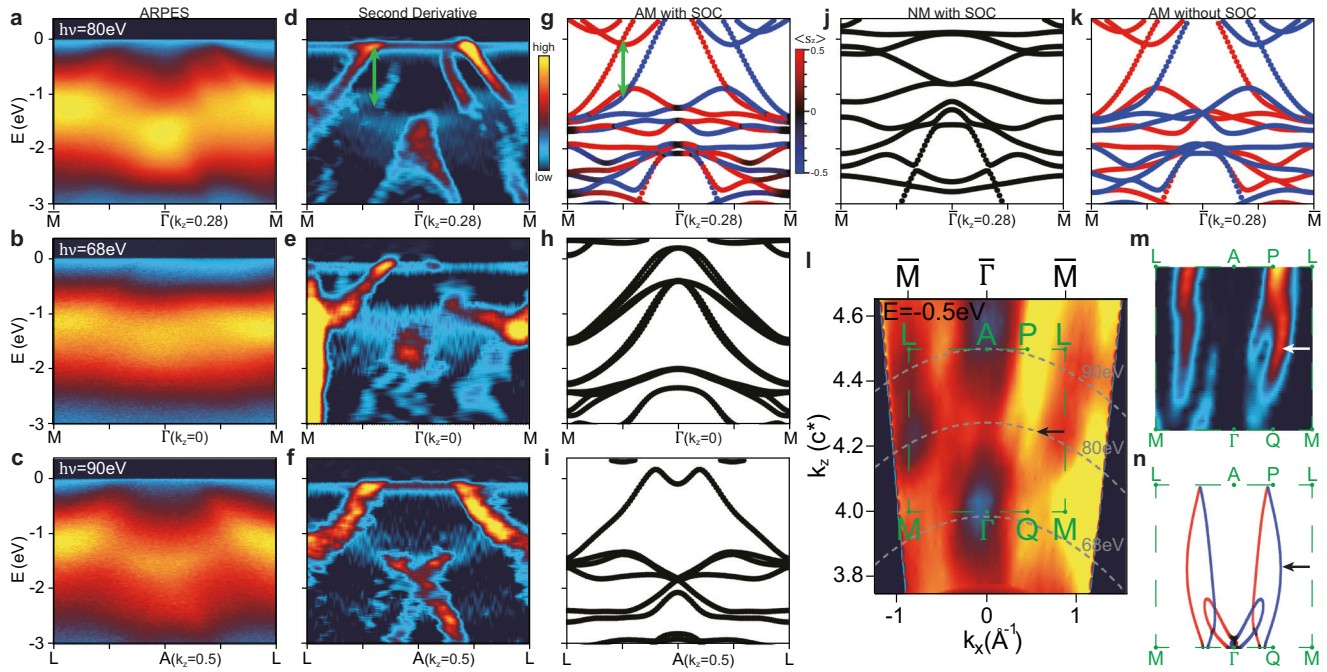

**Fig. 2 | The $k_z$ dependence of alternagnetic splittings along the in-plane $\bar{\Gamma} - \bar{M}$ direction. a–c** ARPES spectra taken with 80 eV (**a**), 68 eV (**b**), 90 eV (**c**) photons. **d–f** The second derivatives of the ARPES spectra corresponding to (**a–c**). **g–i** Spin-resolved band structure from DFT calculations considering both AM and SOC at $k_z = 0.28c^*$ (**g**), 0 (**h**) and $0.5c^*$ (**i**), for comparison with experimental data on the left. The green arrows in (**d, g**) highlight the alternagnetic band splittings. **j, k** DFT calculations at $k_z = 0.28c^*$, considering only SOC (**j**, NM with SOC) or alternagnetism (**k**, AM without SOC), for comparison with (**g**). **l** The $k_x$-$k_z$ map at $E =$ −0.5 eV. The green dashed rectangle marks half of the BZ from $M - \Gamma - M$ ($k_z = 0$) to $L - A - L$ ($k_z = 0.5c^*$). Grey dashed curves indicate the $k_x$-$k_z$ cuts corresponding to (**a–c**). **m** The second derivative of the $k_x$-$k_z$ map within the green dashed rectangle of (**l**). **n** The $k_x$-$k_z$ contour at $E =$ −0.5 eV from DFT, for comparison with (**l, m**). Note that the small feature near $\bar{\Gamma}$ in (**n**) is not observed experimentally. Arrows in (**l–n**) mark the alternagnetic splitting maximized near $k_z = 0.25c^*$. The colors of the curves in DFT calculations indicate the spin polarization [color bar shown in (**g**)].

The in-plane $k_x$-$k_y$ map from ARPES measurements using 200 eV photons is shown in Fig. 1e, which shows (hole-like) pockets centered at $\bar{\Gamma}$ with the in-plane periodicity consistent with the (001) surface. Based on the standard understanding of photoemission[60,61], ARPES measurements under different photon energies probe electronic states with different $k_z$ values. The conversion between photon energy and $k_z$ is determined by the inner potential $V_0$, which is the difference between the crystal potential and the vacuum level. The $k_z$ dispersion of the valence bands along $\bar{\Gamma} - \bar{M}$, obtained from the photon-energy-dependent scan, is shown in Fig. 1f. The observed periodic modulation of the bands with the photon energy allows us to determine the inner potential $V_0$ to be ~17 eV (see also Fig. 2), which is used to convert the photon energy to $k_z$ in Fig. 1f. Note that here the $k_z$ band dispersion is directly related to bulk-type alternagnetic band splitting, according to Fig. 1d. In the ab initio calculations, we find that the alternagnetic splitting in CrSb is maximized along the in-plane $\bar{\Gamma} - \bar{M}$ direction at $k_z \sim 0.25c^*$, with a magnitude of ~1.0 eV. The $k_z$ position where the alternagnetic splitting is maximized is close to the midpoint between two nodal planes at $k_z = 0$ and $0.5c^*$, which is likely related to the high crystal symmetry of CrSb, although such a coincidence is not enforced by any specific symmetry.

## The $k_z$ dependence of bulk-type alternagnetic splitting
The $k_z$-dependent band splitting indicates the bulk-type alternagnetism in CrSb. In Fig. 2a–c, we show the valence bands taken with three representative photon energies (80, 68, and 90 eV) along the $\bar{\Gamma} - \bar{M}$ direction (more data are included in Fig. S8 in[62]). For the 80 eV photons (corresponding to $k_z \sim 0.28c^*$) in Fig. 2a, two sets of hole bands centered at $\bar{\Gamma}$ can be observed with an energy splitting up to 1.0 eV near $E_F$. By contrast, only one hole band can be observed near $E_F$ under 68 eV

(Fig. 2b, $k_z \sim 0$) and 90 eV (Fig. 2c, $k_z \sim 0.5c^*$) photons. These changes in band dispersion with photon energies can be attributed to different $k_z$'s, as discussed above. In addition, the photoexcitation probability for the probed valence bands can also vary with the photon energy and the electron energy/momentum, leading to different photoemission intensity. The valence bands can be better visualized by taking the second derivatives, as shown in Fig. 2d–f. Such a $k_z$-dependent band splitting is consistent with the expected alternagnetic splitting illustrated in Fig. 1d. To make quantitative comparison, we performed ab initio calculations from DFT incorporating both the alternagnetism and SOC (AM with SOC), as shown in Fig. 2g–i. The calculations can well reproduce the experimental band structure, including the hole-type valence bands near $E_F$ and a bundle of bands from −1 eV to −3 eV. Most importantly, the observed $k_z$-dependent splitting near $E_F$ shows good agreement with the ab initio results, as indicated by the green arrows in Fig. 2d, g. The calculations in Fig. 2g–i further show that the split bands at $k_z = 0.28c^*$ exhibit strong spin polarization (red and blue curves), while the $<S_z>$ polarization at the $k_z = 0$ and $k_z = 0.5c^*$ planes is strictly zero (black color), protected by the $[C_2^s\|M_z]$ symmetry.

The alternagnetic order is essential for the observed band splitting, while the SOC plays a relatively minor role. To see this, we present DFT calculations at $k_z = 0.28c^*$ without considering alternagnetism [nonmagnetic (NM) with SOC, Fig. 2j or SOC (AM without SOC, Fig. 2k). Comparing Fig. 2j, k with g, it is clear that the alternagnetism is critical to yield the observed band structure, while the influence from SOC is much weaker. The clear difference between Fig. 2g and j, when comparing with experimental spectra in Fig. 2d, provides direct evidence for the alternagnetic order from the band dispersion.

In general, spin is a good quantum number in alternagnets. When SOC comes into play, the SU(2) symmetry is broken and the bands with

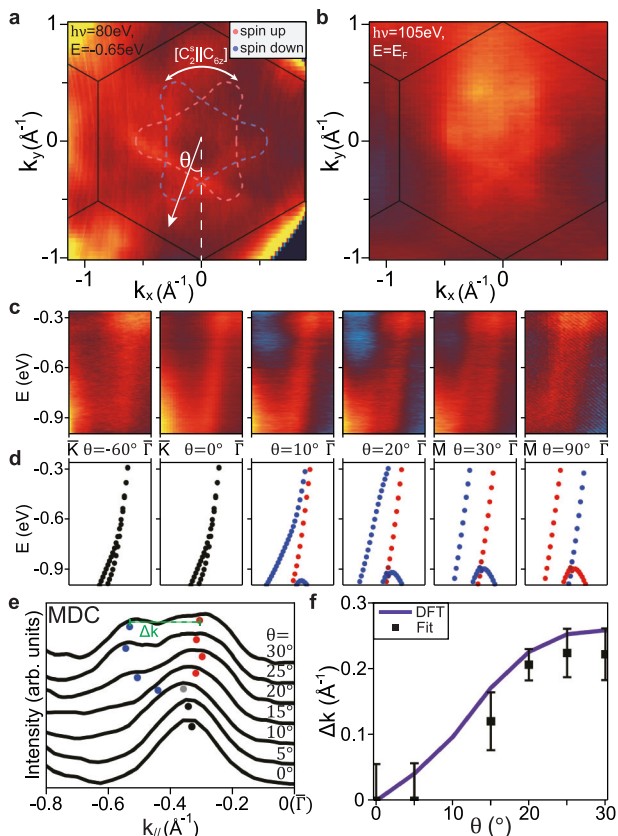

**Fig. 3 | The in-plane alternmagnetic splitting away from the nodal planes. a** The constant-energy $k_x$-$k_y$ map at −0.65 eV, taken with 80 eV photons and using the deflection mode ($k_z$ ~ 0.28$c^*$). The corresponding constant-energy contours from DFT calculations are overlaid on top as dashed curves. **b** The Fermi surface map taken with 105 eV photons, corresponding to $k_z$ ~ 0.2$c^*$. **c** The energy-momentum cuts taken with 80 eV photons for a few representative $\theta$ values [defined in (**a**)]. **d** Band structure from DFT for comparison with (**c**). The momentum range for each plot in (**c**, **d**) is from the BZ boundary (left) to $\bar{\Gamma}$. The curve colors in (**a**, **d**) indicate the spin polarization. **e** MDCs at $E$ = −0.6 eV for $\theta$ from 0° to 30°. Dots indicate fitted peak positions. **f** The extracted alternmagnetic momentum splitting $\Delta k$ (black boxes with error bars) from (**e**), in comparison with DFT calculation (purple curve).

opposite spins will couple to each other, making $S_z$ not a good quantum number. Indeed, if the SOC in CrSb were strong, it would transform the aforementioned nodal planes into an experimentally observable nodal line along the $\Gamma - A$ direction[40]. However, the bands crossing $E_F$ in CrSb are mainly contributed by the 3$d$ orbitals on Cr, which have weak SOC. It is therefore natural to expect that SOC has only small impact over the band structure near $E_F$ (comparing Fig. 2g, k). By calculating the expectation value of $\hat{S}_z$, as shown in Fig. 2g, we find that the spin quantum number near $E_F$ can indeed be approximated as a half-integer ($\pm\frac{1}{2}$). However, far below $E_F$ where appreciable contributions from Sb 5$p$ orbitals are present, e.g., $E$~−2 eV, the inclusion of SOC makes $< S_z >$ deviate from $\pm 1/2$ appreciably and leads to noticeable in-plane spin polarization (see Fig. S12 in[62]). Another effect from SOC is the tiny band splitting at $k_z$ = 0 and concomitant non-zero in-plane spin components $< S_x >$ and $< S_y >$ (see Fig. S7 and Fig. S12 in[62]), which is called weak alternmagnetism in[13]. Such small splitting cannot be resolved in our experiments (comparing Fig. 2e, h).

The $k_x$-$k_z$ map at $E$ = −0.5 eV with fine $k_z$ steps is shown in Fig. 2l; its second derivative and the calculated contour from DFT are displayed in Fig. 2m and n, respectively. Along the $P - Q$ momentum direction as defined in[51] (also labelled in Fig. 2l–n), the raw data already show a double-string feature (see arrow in Fig. 2l) with closed ends at both $P$ and $Q$ points, which can be better identified from the second derivative

in Fig. 2m. The experimental results and DFT calculations agree quite well: the alternmagnetic splitting, i.e., the separation between the two strings, is close to zero at $k_z$ = 0 and gradually increases to its maximal value at $k_z$~0.25$c^*$ (marked by arrows in Fig. 2l–n), eventually decreasing to zero at $k_z$ = 0.5$c^*$. Our observed splitting along the $P - Q$ direction is overall consistent with the previous ARPES study from (100)-oriented CrSb films[51], although the measurement geometries are different. Our $k_z$-dependent study unambiguously confirms the bulk-type alternmagnetic splitting in CrSb.

### The in-plane symmetry of *g*-wave alternmagnetism

The in-plane momentum dependence of the band splitting provides direct evidence for the *g*-wave alternmagnetism in CrSb. Figure 3a shows the $k_x$-$k_y$ map at $E$ = −0.65 eV taken with 80 eV photons ($k_z$ ~ 0.28$c^*$), where the flower-like hole-type pockets centered at $\bar{\Gamma}$ can be observed. The constant-energy contours from ab initio calculations are overlaid on top in Fig. 3a, which agree well with the experimental data. Based on the spin group symmetry analysis, the spin-resolved contours have three-fold rotational symmetry, and the bands with different spins are related by a 60° rotation, leading to the observed six-fold flower-like pockets in the spin-integrated ARPES. Since the photoemission intensity near $E_F$ is very low under 80 eV photons, we further show the Fermi surface map taken with 105 eV photons in Fig. 3b, which corresponds to $k_z$ ~ 0.2$c^*$. Here the six-fold flower-like pockets expected from DFT calculations are clearly visible from the raw data, highlighting the large alternmagnetic splitting right at $E_F$. Figure 3c shows energy-momentum cuts taken with 80 eV photons at a series of representative $\theta$ angles, where $\theta$ is the azimuthal angle between the momentum **k** and the $\bar{\Gamma} - \bar{K}$ direction, defined in Fig. 3a. The results clearly demonstrate the presence (absence) of alternmagnetic band splitting along $\bar{\Gamma} - \bar{M}$ ($\bar{\Gamma} - \bar{K}$), as well as the gradual evolution of the splitting between these two directions. These experimental results are consistent with the corresponding spin-resolved band structures from ab initio calculations in Fig. 3d (see Fig. S9 in[62] for more details).

To quantitatively analyze the in-plane dependence of alternmagnetic splitting, we also extract the momentum distribution curves (MDCs) at $E$ = −0.6 eV from Fig. 3c and plot them in Fig. 3e. The MDCs are then fitted with two Gaussian peaks (see Fig. S10 in[62] for details) to yield the momentum splitting ($\Delta k$) as a function of $\theta$ (Fig. 3f). The momentum splitting agrees well with the ab initio calculation (see also Fig. S10 in ref. [62]), including the $|\sin(3\theta)|$-like functional form and the magnitude of splitting, although small deviations are likely present due to the experimental uncertainty or inaccuracy in calculations. We mention that the bulk-type *g*-wave alternmagnetism only requires a six-fold in-plane symmetry in the magnitude of the band splitting, although the exact location of the nodal planes (along either $\bar{\Gamma} - \bar{K}$ or $\bar{\Gamma} - \bar{M}$) is still dependent on the details of the lattice[9].

### Evidence of spin polarization from spin-resolved ARPES

To verify the spin polarization of the alternmagnetically split bands, separate spin-resolved ARPES measurements were performed along the $\bar{\Gamma} - \bar{M}$ direction with 80 eV photons ($k_z$ = 0.28$c^*$), where the splitting is nearly maximized. Before spin-resolved measurements, the spin-integrated spectrum is checked first (Fig. 4a), and its second derivative (Fig. 4b) shows two sets of hole pockets near $\bar{\Gamma}$ from alternmagnetism, similar to those in Fig. 2a, d. The different data quality in Figs. 4a and 2a can be attributed to different experimental conditions.

Figure 4 c shows the spin-resolved MDCs taken along the white dashed cut in Fig. 4a, b. Here the spin detector measures the out-of-plane spin component ($S_z$), and the spin-up (spin-down) MDC is shown as red (blue) curve, respectively. It is clear that both spin-up and spin-down MDCs exhibit a double-peak structure with large broadening, corresponding to the spin-up and spin-down bands labelled in Fig. 4b. The obvious shifting of the peak center from spin-up to spin-down

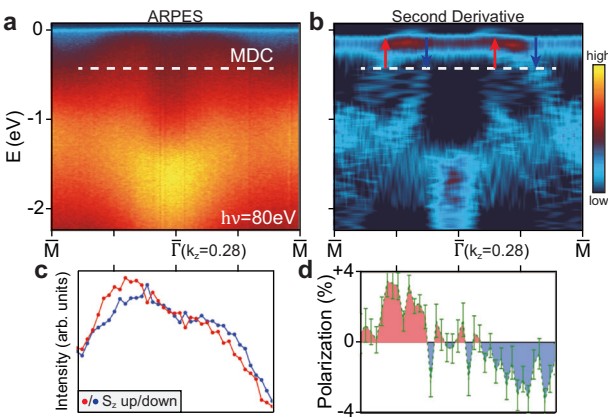

**Fig. 4 | Evidence of spin polarization from spin-resolved ARPES measurements. a** Spin-integrated ARPES spectrum along $\bar{\Gamma} - \bar{M}$ taken with 80 eV photons, for the sample used for spin ARPES measurements. **b** The second derivative of (**a**). The expected spin polarizations of the split bands near $E_F$ are labelled by red (spin-up) and blue (spin-down) arrows. **c** Spin-resolved MDCs along the white dashed cut in (**a**, **b**). The red (blue) curve indicates the spin-up (spin-down) MDC for the spin component along $z$, i.e., $S_z$. **d** The spin polarizations extracted from the spin-resolved MDCs in (**c**). Error bars are defined as the standard error of polarization at each $k$ point between repeated scans.

MDCs directly reflects the spin splittings expected from the altermagnetism in CrSb. Based on the calibrated Sherman function from Au(111), the corresponding spin polarization is extracted and shown in Fig. 4d. Ideally, for a single altermagnetic domain with small momentum broadening, the spin polarization along this momentum cut would be nearly 100% and exhibit a up-down-up-down pattern (or down-up-down-up, depending on the domain orientation), according to DFT calculations shown in Fig. 2g. However, the spin polarization extracted experimentally is much smaller (Fig. 4d), particularly for the inner hole pocket. There are two possible reasons for the reduced spin polarization: first, the probed area likely contains two domains with opposite spin splittings, leading to significant reduction of spin polarization, which was also observed in MnTe[13]. Note that the mixing of different domains in CrSb does not affect the spin-integrated ARPES spectra, due to the Néel vector aligning along the $c$ axis, but the in-plane oriented Néel vector in MnTe can lead to different spin-integrated ARPES spectra for different domains[13]. Second, the large momentum broadening in the ARPES data can further reduce the spin polarization due to overlap of opposite-spin bands. Nevertheless, our direct observation of appreciable spin polarizations in the altermagnetically split bands and their consistencies with the DFT calculations provide strong support for the altermagnetic spin splittings in CrSb.

**Mechanism behind the large alter magnetic splitting**

Although the altermagnetic band splitting can be characterized by its corresponding spin group symmetry, its amplitude is dependent on the chemical bonding strength in real material. In this section, we construct a microscopic TB model to delve into the mechanism behind the large altermagnetic splitting in CrSb. Based on orbital analysis of the electronic band structure of CrSb, we find that the set of bands intersecting $E_F$ is primarily contributed by the $d_{xz}$, $d_{yz}$, $d_{xy}$, and $d_{x^2-y^2}$ orbitals on Cr, which are hybridized with the $p$ orbitals on Sb (Details can be found in Section 1 of [62]). It is important to note that both ($d_{xz}$, $d_{yz}$) and ($d_{xy}$, $d_{x^2-y^2}$) share the same symmetry property (the irreducible representation $E_g$) under the site symmetry of Cr (point group $D_{3d}$). Therefore, one $E_g$ per Cr atom is sufficient to construct the effective model. A comprehensive derivation of the TB model is provided in Section 1 of [62]. Ignoring SOC, the model can be formally expressed as

follows,

$$H(\mathbf{k}) = \begin{bmatrix} H^{\uparrow}(\mathbf{k}) & 0_{4\times4} \\ 0_{4\times4} & H^{\downarrow}(\mathbf{k}) \end{bmatrix} = \sum_{i=0,3} \sum_{j,m=0,1,2,3} f_{ijm}(\mathbf{k})\Gamma_{ijm} \quad (1)$$

where $\Gamma_{ijm} = s_i \otimes \sigma_j \otimes \tau_m \equiv s_i\sigma_j\tau_m$. The Pauli matrices $s_{i=0,1,2,3}$ (or $\sigma_i$ or $\tau_i$) are used here to represent the space of spin (or the two sublattices $Cr_1$ and $Cr_2$ or the two orbitals $\phi_1$ and $\phi_2$ about $E_g$).

By incorporating all the spin group symmetries and considering the kinetic hopping terms up to the 3NN sites, we have found that both the NN and the NNN hopping terms preserve the translational spin-flip operation $[C_2^s \| \vec{t}]$[62]. It indicates that neither the NN nor the NNN hopping results in an altermagnetic spin splitting (see Fig. 1a for illustration of NN, NNN and 3NN hoppings). It is only when we take into consideration the 3NN terms that the altermagnetic splitting occurs. As demonstrated in[62], there are five 3NN terms in Eq. (1), i.e., $\Gamma_{011}$, $\Gamma_{013}$, $\Gamma_{010}$, $\Gamma_{311}$ and $\Gamma_{313}$, and their coefficients are

$$f_{011}(\mathbf{k}) = 2\sqrt{3}(q_4 - q_1)\sin^2\left(\frac{k_1}{2}\right)\cos\left(\frac{k_3}{2}\right),$$

$$f_{013}(\mathbf{k}) = 2(q_4 - q_1)\sin^2\left(\frac{k_1}{2}\right)\cos\left(\frac{k_3}{2}\right),$$

$$f'_{010}(\mathbf{k}) = 2(q_1 + q_4)\cos\left(\frac{k_3}{2}\right)[2\cos(k_1) + 1], \quad (2)$$

$$f_{311}(\mathbf{k}) = 3(q_2 + q_3)\sin(k_1)\sin\left(\frac{k_3}{2}\right),$$

$$f_{313}(\mathbf{k}) = \sqrt{3}(q_2 + q_3)\sin(k_1)\sin\left(\frac{k_3}{2}\right),$$

where $q_i$'s ($i = 1, 2, 3, 4$) are the independent 3NN hopping parameters between Cr atoms at $R_1 = (0, 0, 0)$ and $R_2 = (1, 1, \frac{1}{2})$, as defined below[62]

$$\begin{aligned} q_1 &= \langle \phi_1, Cr_1, R_1 | \hat{H} | \phi_1, Cr_2, R_2 \rangle, \\ q_2 &= \langle \phi_1, Cr_1, R_1 | \hat{H} | \phi_2, Cr_2, R_2 \rangle, \\ q_3 &= \langle \phi_2, Cr_1, R_1 | \hat{H} | \phi_1, Cr_2, R_2 \rangle, \\ q_4 &= \langle \phi_2, Cr_1, R_1 | \hat{H} | \phi_2, Cr_2, R_2 \rangle. \end{aligned} \quad (3)$$

The first three 3NN terms in Eq. (2), i.e., $f_{011}(\mathbf{k})\Gamma_{011}$, $f_{013}(\mathbf{k})\Gamma_{013}$ and $f'_{010}(\mathbf{k})\Gamma_{010}$, preserve the $[C_2^s \| \vec{t}]$ symmetry, and they do not give rise to any altermagnetic splitting. In contrast, the last two 3NN terms $f_{311}(\mathbf{k})$ $\Gamma_{311}$ and $f_{313}(\mathbf{k})\Gamma_{313}$, which are proportional to $q_2 + q_3$, break both $[T\|P]$ and $[C_2^s \| \vec{t}]$ symmetries, leading to the altermagnetic band/spin splitting.

By fitting the band structure from DFT calculations, especially along the $\bar{\Gamma} - \bar{M}$ direction near $E_F$, we obtain all the free parameters used in the TB model (the resulting band dispersion of the TB model is shown in Fig. 5b). We find that the fitted 3NN hopping parameters, especially $q_2 + q_3$ that gives rise to the spin splitting, are of the similar magnitude as the NNN hoppings (see Table 2 in[62]), which is counterintuitive. Indeed, $d$ orbitals are usually very local, and the direct 3NN hopping should be very small. This is further confirmed by the DFT calculations, as shown in Table 1. In the section 2 of[62], our additional analysis based on the Wannier calculations show that the strong 3NN hoppings are dominated by a second-order hopping process mediated by Sb. In other words, the $p$ orbitals on Sb assist the 3NN hopping between $d$ orbitals on Cr. As tabulated in Table 1, the amplitude of $q_2 + q_3$ that arises from the direct (DH) and assistant hopping (AH) processes are about 21 and 152 meV, respectively. Their total contribution (TH) is indeed very close to the one used in the TB model. Therefore, the Sb ions between the two Cr sublattices play a crucial role for the surprisingly large 3NN hopping and hence the strong altermagnetic splitting in CrSb. These results imply that the

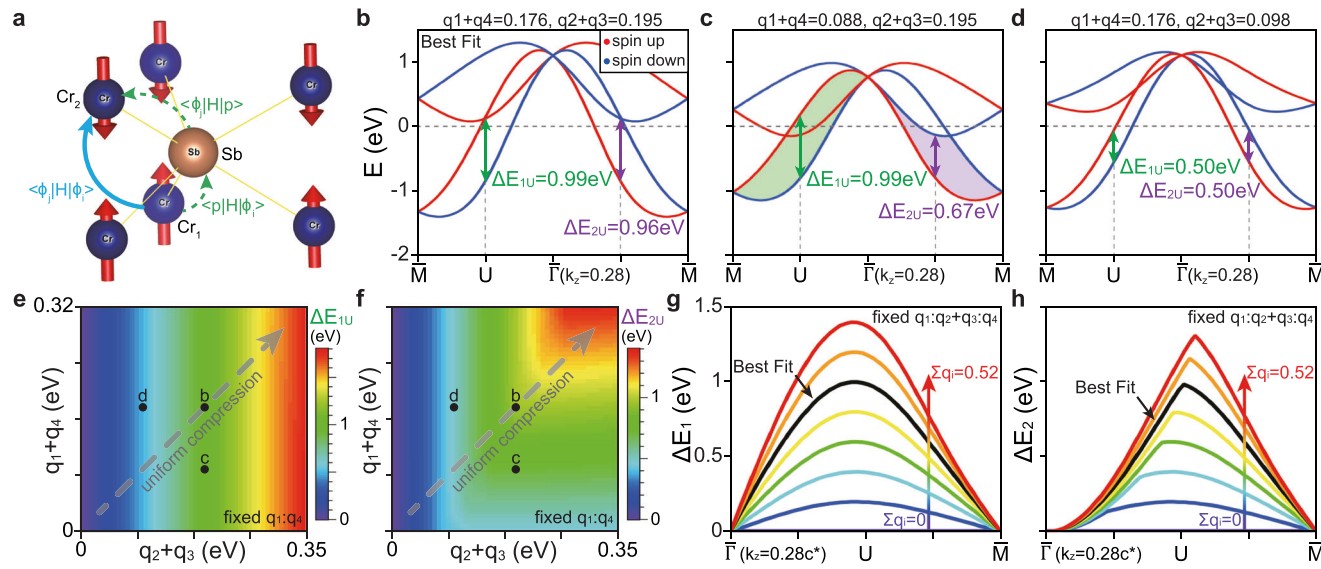

**Fig. 5 | Origin of the large alternmagnetic splitting in CrSb. a** The dominant hopping process between the 3NN Cr atoms at $R_1 = (0, 0, 0)$ (Cr$_1$) and $R_2 = (1, 1, \frac{1}{2})$ (Cr$_2$), which is mediated by the $p$ orbitals on the Sb atom at $(\frac{1}{3}, \frac{2}{3}, \frac{1}{4})$. Here we use Cr$_1$ (Cr$_2$) to present the Cr sublattice with up (down) spins. **b** Band dispersions along the $\bar{\Gamma} - \bar{M}$ direction at $k_z = 0.28c^*$ obtained from the eight-band TB fit to DFT calculation. The red and blue lines represent spin-up and spin-down bands, respectively. The alternmagnetic splitting within one band ($\Delta E_{1U}$) and the splitting of

opposite-spin bands closest to $E_F$ ($\Delta E_{2U}$) at the $U$ point are both defined. **c, d** Band dispersions with $q_1 + q_4$ (**c**) or $q_2 + q_3$ (**d**) reducing to half of its value in (**b**). **e, f** $\Delta E_{1U}$ (**e**) and $\Delta E_{2U}$ (**f**) as a function of $q_2 + q_3$ and $q_1 + q_4$. The black dots label the positions corresponding to (**b**–**d**). The color bars for the image plots are shown on the right. **g, h** $\Delta E_1$ (**g**) and $\Delta E_2$ (**h**) along the $\bar{\Gamma} - \bar{M}$ direction at $k_z = 0.28 c^*$ by increasing the value of $\sum q_i$ from 0 to 0.52 eV, where the ratio $q_1 : q_2 + q_3 : q_4$ is fixed. The black curve corresponds to the best fit to DFT (**b**).

alternmagnetic splitting in CrSb can be tuned by pressure or strain, which can strongly modify the hopping processes assisted by Sb ions.

The band dispersions with $q_1 + q_4$ or $q_2 + q_3$ values reduced to half of the values in Fig. 5b are displayed in Fig. 5c, d, respectively. These results show that while $q_1 + q_4$ controls the overall bandwidth, $q_2 + q_3$ dictates the alternmagnetic splitting within one set of band. Due to the multiorbital nature of CrSb, it is useful to define two types of spin splittings: the alternmagnetic splitting within one set of band as $\Delta E_1$ and the splitting of opposite-spin bands closest to $E_F$ as $\Delta E_2$ - their values at the $U$ point, i.e., the midpoint between $\bar{\Gamma}$ and $\bar{M}$ in the $k_z = 0.28 c^*$ plane, are labelled in Fig. 5b–d. While $\Delta E_1$ has a simple physical meaning, $\Delta E_2$ is directly relevant to the spin polarization near $E_F$ (and hence the spintronic applications). In Fig. 5e, f, we systematically investigate the evolution of $\Delta E_{1U}$ and $\Delta E_{2U}$ as a function of $q_2 + q_3$ and $q_1 + q_4$. Our results show that $\Delta E_{1U}$ is approximately proportional to the magnitude of $q_2 + q_3$ and is almost independent of $q_1 + q_4$ in this parameter region, as expected by our TB model. On the other hand, $\Delta E_{2U}$ becomes dependent on $q_1 + q_4$ when $q_2 + q_3$ becomes larger than ~ 0.1 eV. This is because when $q_2 + q_3$ becomes large, the top conduction bands with similar spin splitting will now cross $E_F$ (see Fig. 5c), reducing $\Delta E_{2U}$ accordingly. This implies that for a multiorbital system like CrSb, a careful tuning of different band parameters is necessary to generate maximal spin splitting near $E_F$. Figure 5g, h show the evolution of $\Delta E_1$

and $\Delta E_2$ along $\bar{\Gamma} - \bar{M}$ at $k_z = 0.28 c^*$, by increasing the numerical values of $\{q_i\}$ while keeping a fixed ratio among them (corresponding to the grey dashed lines in Fig. 5e, f). These calculations, which can simulate the process of uniform compression, indicate a clear and monotonic increase in the magnitude of alternmagnetic splitting under hydrostatic pressure.

## Discussion

It is interesting to compare CrSb with MnTe, another bulk-type $g$-wave alternmagnet with similar crystal structure[13]. There are several important differences between these two systems. Firstly, the 3$d$ spins in CrSb are aligned along the $c$ axis, in contrast to the in-plane 3$d$ spins in MnTe. This eliminates multiple band splittings caused by the different in-plane domains as reported in MnTe[13,15,63] and allows for clear identification of alternmagnetic splitting in CrSb. Secondly, the valences of Cr and Mn should be different, leading to different fillings of the 3$d$ valence bands and hence distinct ground states (metallic in CrSb and semiconducting in MnTe). Finally, the lattice constant $c$ in CrSb (5.44 Å) is considerably smaller than that of MnTe (6.75 Å), which, together with the stronger electron correlation in MnTe with $U$ ~4.8 eV[13], gives rise to stronger electron hopping and hence larger alternmagnetic splitting up to 1.0 eV near $E_F$ in CrSb. By contrast, the observed alternmagnetic splitting in MnTe is $\lesssim 0.5$ eV and is located far below $E_F$, which is mainly derived from Te $p$ orbitals[12,13]. We note that our TB model might not be directly applicable to MnTe, due to different orbital character and nature of magnetism.

The mechanism of large alternmagnetic splitting in CrSb can be further illustrated by comparison with VNb$_3$S$_6$, which is another proposed bulk-type $g$-wave alternmagnet with the identical spin group symmetry as CrSb and MnTe[10]. Since the magnetic moments of V are antiparallel between two neighboring layers, the alternmagnetic splitting in VNb$_3$S$_6$ also originates from the interlayer hopping. Yet its much larger lattice constant $c$ = 12.17 Å[64] and weak interlayer coupling between V and NbS$_2$ layers result in a very small splitting of a few tens of meV (see Fig. S11 in[62]).

## Table 1 | Values of the 3NN hopping parameters

| Parameter | $q_1$ | $q_2 + q_3$ | $q_4$ |
|---|---|---|---|
| TB | −0.0184 | 0.1950 | 0.1942 |
| DFT(DH) | 0.0050 | 0.0214 | −0.0660 |
| DFT(AH) | −0.0234 | 0.1516 | 0.2602 |
| DFT(TH) | −0.0184 | 0.1730 | 0.1942 |

The parameters in the second row are the ones used in the fitted eight-band TB model. The parameters in the third and fourth rows indicate contributions of the direct hopping (DH) and assistant hopping (AH) processes obtained from DFT calculations. The last row is the total hopping (TH) after summation of the direct and assistant hoppings. The energy unit is eV.

The large bulk-type $g$-wave altermagnetic splitting at $E_F$ in CrSb provides interesting opportunities to explore novel superconducting proximity effects at the interfaces between metallic altermagnets and conventional superconductors. For instance, it has been theoretically proposed that the interface between a conventional superconductor and a planar $d$-wave altermagnet can induce finite-momentum pairing[44], and their planar Josephson junctions can exhibit orientation-dependent 0-$\pi$ oscillations[45]. In contrast, due to the three-dimensional spin-degenerate nodal planes of CrSb, both planar and vertical Josephson junctions composed of CrSb could exhibit similar (or even richer) superconducting properties. For example, the vertical Josephson junction could achieve damped oscillating pairing along any direction non-parallel to the nodal planes, facilitating the toggling of finite-momentum pairing on and off by controlling the orientation of the interface. Therefore, our work will motivate future theoretical and experimental studies to explore such intriguing phenomena.

In summary, we present direct spectroscopic evidence of bulk-type $g$-wave altermagnetism in CrSb from high-resolution synchro-tron-based ARPES and spin-resolved ARPES measurements. Our systematic $k_z$ and in-plane momentum dependent study confirms a large altermagnetic splitting up to 1.0 eV near $E_F$, which is in good agreements with the DFT calculations. The spin polarization of split bands is further verified by spin-resolved ARPES measurements. Our TB model analysis reveals that the altermagnetic splitting arises mainly from the 3NN hopping of Cr $3d$ electrons mediated by Sb $5p$ orbitals, highlighting the important role of electronic coupling between magnetic and nonmagnetic ions. The insight can be important for finding/tuning altermagnetic materials with enhanced properties. The large altermagnetic splitting near $E_F$, together with the high ordering temperature up to 700 K in CrSb, paves the way for exploring emergent phenomena and practical applications based on altermagnets.

Note: during the review process of this paper, we became aware of other independent ARPES works on CrSb posted on arXiv[65–67], which also support the altermagnetism in CrSb.

## Methods

### Single crystal growth and characterizations
Single crystals of CrSb were grown using an antimony flux. The elements were combined in a molar ratio of Cr:Sb of 2:3, and sealed in an evacuated quartz tube. The tube was heated up to 1150 °C and held at this temperature for 20 h, before being cooled down slowly to 700 °C and centrifuged to remove the excess antimony flux. The lateral size of the single crystals is typically $1 \times 1$ mm$^2$. The chemical composition of the crystals was confirmed by energy dispersive X-ray spectroscopy. The crystal structure was checked by XRD measurements (Fig. 1b and Fig. S5b in[62]). The temperature-dependent resistivity was measured using a Physical Property Measurement System (PPMS), and it shows metallic behavior with no phase transition from 300 K down to 1.8 K (Fig. S5a in[62]), consistent with previous reports[68]. The cleaved single crystals were also characterized by core-level photoemission measurements, which show expected core level peaks from Cr $3p$ and Sb $4d$ electrons (Fig. S5c in[62]), without any trace of impurities.

### Synchrotron-based ARPES measurements
The spin-integrated ARPES measurements were carried out at the BLOCH beamline in Max IV Lab (Sweden) and the BL03U beamline at Shanghai Synchrotron Radiation Facility (SSRF, China). The typical energy resolution was set to ~20 meV and the measurement temperature is ~20 K. The typical beam spot is ~$15 \times 15\ \mu$m$^2$ for the BLOCH beamline and ~$50 \times 50\ \mu$m$^2$ for the BL03U beamline. The ARPES measurements were performed on the (001)-oriented surface, which was cleaved $in-situ$ at measurement temperature and under ultrahigh vacuum (<$5 \times 10^{-11}$ mbar). All ARPES data in our manuscript were taken with horizontally polarized photons.

The spin-resolved ARPES measurements were conducted at the APE-LE beamline in Elettra Sincrotrone Trieste (Italy). The typical energy and angular resolutions are ~100 meV and ~1.5°, respectively. The measurement temperature is ~15 K. The typical beam spot is ~$150 \times 50\ \mu$m$^2$. The Sherman function of the spin detector is 0.3, calibrated by the Rashba-split surface states from Au(111) right before the experiment. The spin-resolved ARPES measurements were achieved through VESPA (Very Efficient Spin Polarization Analysis), which consists of two very low energy electron diffraction (VLEED)-based scattering chambers and is able to measure all three spin components of the photo-emitted electrons[69].

### Ab initio calculations
The DFT calculations were performed using the Vienna Ab initio Simulation Package (VASP)[70,71]. The Perdew, Burke and Ernzerhoff (PBE) parametrization of generalized gradient approximation (GGA) to the exchange-correlation functional was employed[72]. We performed calculations with different choices of the coulomb repulsion $U$ (see Fig. S6 in[62]), and we found that $U = 0$ shows the best agreement with the experimental results. Therefore, $U$ is set to zero in all calculations, unless noted otherwise. An energy cutoff of 345 eV and $11 \times 1 \times 9$ Γ-centered K-mesh were employed to converge the total energy to 1 meV/atom. The band structure obtained from PBE method were fitted to a tight-binding model Hamiltonian with maximally projected Wannier function method, which was then used to calculate the Fermi surface with wannier_tools package. To understand the underlying origin of altermagnetic splitting, calculations with either SOC or AM turned off, i.e., AM without SOC or NM with SOC, were both performed (see Fig. S7 in[62]). The results show that altermagnetism plays a dominant role in the observed band structure, while the SOC effect is not very important.

## Data availability
All the source data accompanying this manuscript have been deposited in the Figshare repository, accessible via https://doi.org/10.6084/m9.figshare.25865749.

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

## Acknowledgements

This work is supported by the National Key R&D Program of China (Grant No. 2023YFA1406303, No. 2022YFA1402200), the Key R&D Program of Zhejiang Province, China (2021C01002), the National Natural Science Foundation of China (No. 12174331, No. 12204159, No. 12374163, No. 12350710785) and the State Key project of Zhejiang Province (No. LZ22A040007). We acknowledge MAX IV Laboratory for time on Beamline BLOCH under Proposal 20230343. Research conducted at MAX IV, a Swedish national user facility, is supported by the Swedish Research council under contract 2018-07152, the Swedish Governmental Agency for Innovation Systems under contract 2018-04969, and Formas under contract 2019-02496. We also acknowledge Elettra Sincrotrone Trieste for time on Beamline APE-LE under Proposal 20240269. This work has been partly performed in the framework of the nanoscience foundry and fine analysis (NFFA-MUR Italy Progetti Internazionali) facility. Part of this research used beamline 03U of the Shanghai Synchrotron Radiation Facility, which is supported by the ME2 project under Contract No. 11227902 from the National Natural Science Foundation of China. We would like to thank Prof. Dawei Shen, Dr. Zhengtai Liu, Dr. Balasubramanian Thiagarajan, Dr. Mats Leandersson, Dr. Indrani Kar, Ms. Jiayi Lu and Prof. Guanghan Cao for experimental help. We are also grateful to Prof. Yulin Chen, Prof. Chang Liu, Mr. Qi Jiang and Prof. Xin Lu for useful discussions. Some of the images in the paper were created using VESTA software[73].

## Author contributions

The project was designed by Y.L. Single crystal growth and characterization was done by S.Y., J.L., J.Z., H.Y. and Y.Z., with help from S.C. and Y.S. The ARPES measurement was performed by G.Y., with help from H.Z., W.Z., Z.P., Yifu Xu, A.J., J.F., I.V., W.Z., M.Y., L. Y., M.S. and Y.L. ARPES data analysis was done by G.Y. and Y.L., with help from Z. L. and Yuanfeng Xu. Ab initio calculations and tight-binding model analysis were carried out by Z.L. and Yuanfeng Xu. L.-H. H. and M.S. contribute to the discussion. The manuscript was prepared by G.Y., Z.L., Yuanfeng Xu and Y. L., with inputs from all other coauthors. All authors discussed the results and commented on the manuscript.

## Competing interests

The authors declare no competing interests.
