## [Transparent Peer Review file · Nature Communications]

Three-dimensional mapping of the altermagnetic spin splitting in CrSb

Corresponding Author: Professor Yang Liu

Version 0:

Reviewer comments:

Reviewer #1

(Remarks to the Author)

Attached

Reviewer #2

(Remarks to the Author)

The authors have studied one of the altermagnet candidates, CrSb, using synchrotron-based ARPES. Compared to previous experimental work on CrSb thin films [Ref.41], this study resolves spin-splitting more clearly, with a maximum separation of 1 eV. They also revealed g-type spin polarization by comparing their results with ab initio calculations. I believe this work merits publication in Nature Communications, provided the following concerns are addressed. All of my concerns stem from the poor resolution in some of the experimental results, which currently hinders a clear conclusion regarding the match between theoretical and experimental findings.

1. Although the split feature is recognizable in Fig. 2a & d ($h\nu=80$ eV), its visibility in the constant energy contour is very poor. Despite the authors' optimistic statement that "The experimental results and DFT calculations agree quite well" (p.4), the string-like feature with closed points at both ends is hardly visible. I suggest creating a second derivative image to improve visibility

2. The same issue occurs in the constant energy surfaces in Fig. 3a & b. The pair of deformed triangles (dashed lines) seen in the calculations cannot be observed in the experimental results. This part is crucial for supporting the claim of g-wave type spin texture, but it is currently unconvincing.

To address the poor resolution in the constant energy contours, it might be worthwhile to perform spin-resolved ARPES to confirm the type of spin-polarization. However, this could be challenging due to the mixture of different magnetic domains. A previous experiment on MnTe attempted this but failed to resolve it well, resulting in only a tiny spin polarization [Ref.12]. As the authors claim in this manuscript (p.6) "Firstly, the 3d spins in CrSb are aligned along the c axis, in contrast to the in-plane 3d spins in MnTe. This eliminates multiple band splittings caused by the different in-plane domains as reported in MnTe [12,14,52] and allows for clear identification of altermagnetic splitting in CrSb.", clearer spin polarizations are highly expected for CrSb.

I wonder if the authors have already attempted such measurements. If so, I recommend including some comments on the spin polarization measurements.

Reviewer #3

(Remarks to the Author)

The article provides a three-dimensional k-space mapping of the CrSb. However, I have some criticism that should be addressed by the authors before publication: As a summary, the article contains valuable information about these CrSb. Especially remains challenging in the three-dimensional k-space mapping. Their interesting results are well captured

between experiments and theories. My opinion of the paper is quite positive, but I would like some moderate revisions before acceptance.

Below, you find my questions/remarks:

- 1, In ab initio calculations, the antiferromagnetic splitting of CrSb is maximized in the in-plane $\Gamma - M$ direction with $k_z \sim 0.25c$, what is the reason for this result? And why this trend?
- 2, In Fig2a-c, the mechanism of valence band changes at three different photon energies should be explained?
- 3, In Fig3a-b, could similar flower-like hole-type pockets be observed at photon energies other than 80 eV photons? I doubt it.
- 4, At $q_2 + q_3$ greater than about 0.2 eV, the effects of q_1 and q_4 become significant; what is the physical mechanism for this transition? In addition, what other factors besides the 3NN hopping may have an effect on the antiferromagnetic splitting?
- 5, What is the matrix representation of symmetric operations?
- 6, There is a clear $\{C_z \mid \infty\} \mid E$ symmetry operation regarding spin in the structure, which is the infinitesimal rotation of spin about the z-axis that the author did not consider.
- 7, The format of the references is extremely poor, with many of them having inconsistent formats and missing some theoretical predictions, especially the recently published such as APL, et.al

Version 1:

Reviewer comments:

Reviewer #1

(Remarks to the Author)
Attached

Reviewer #2

(Remarks to the Author)

I would like to express my deep appreciation for the authors' tremendous efforts in conducting another experiment to detect the spin polarization of the split energy bands, which appears to align well with theoretical expectations. Although the magnitude of the spin polarization is smaller than the predicted value, it is reasonable and sufficient to support their claim of observing spin-split antiferromagnetic bands. The newly added Fermi surface contour measured at 105 eV is commendable as well. I believe they have done their utmost to detect spin polarization.

I understand that such photoemission experiments cannot be performed under strong external magnetic fields, in contrast to magnetic circular dichroism experiments. Additionally, obtaining a single magnetic domain for antiferromagnetic materials poses challenges, as spin-resolved photoemission requires, in principle, a single "remanent" magnetic domain.

I find that the authors have addressed all concerns raised by the three referees, including myself, in a reasonable manner. Therefore, I recommend this work for publication in Nature Communications.

Reviewer #3

(Remarks to the Author)

The author has responded to each of the questions I raised, and I think they can now be accepted by NC.

Re: NCOMMS-24-30428

Authors: Guowei Yang et al.

Dear reviewers,

Thank you very much for your constructive comments on our manuscript, which help us improve the manuscript. Following your suggestions, we now include in our revised paper additional experimental data, especially high-resolution maps and spin-resolved ARPES results, as well as more calculations and analysis. The manuscript and SI are revised accordingly.

Below we addressed the reviewers' questions in a point-by-point fashion (All the revised parts in the manuscript and SI are highlighted in red).

Reviewer #1 (Remarks to the Author):

CrSb is a representative altermagnet in the emerging class of collinear magnets. The manuscript by G. Yang and coworkers reports on the experimental and theoretical aspects of the CrSb altermagnet. This is an interesting follow-up study of the pioneering work by S. Reimers et al. (Ref.41) because:

- (i) the ARPES measurements were carried out at different locations of the CrSb Brillouin zone
- (ii) the authors compare their experimental results within the context of tight-binding model, thereby discussing the CrSb altermagnetic splitting by estimating the role of the mediation of the nearest neighbor Sb ions

Overall, the manuscript is a valuable contribution to characterizing the CrSb altermagnetism. However, this reviewer is not convinced that the manuscript is suitable for publication in Nature Communications. Several issues preclude publication in the current form. Here is a summary of the main issues I'd like to put forward for the general audience interested in CrSb altermagnetism:

We thank the reviewer for appreciating the contribution of our paper to the community. We are also grateful for the useful comments. Here we would like to clarify one point: while the pioneering work by S. Reimers (old ref. 41, now ref. 51) reported experimental observation of altermagnetic splitting in CrSb, high-resolution three-dimensional (3D) mapping of altermagnetic splitting and direct observation of spin polarization are still lacking. Our work includes a 3D mapping of the altermagnetic splitting, both along k_z (Fig. 2) and in the k_x - k_y plane (Fig. 3), which is important to establish the bulk-type g-wave altermagnetism in CrSb. The newly added spin-ARPES results (Fig. 4) provide direct support for the spin polarization of the altermagnetically split bands.

A) In the field of altermagnetism, there is an ongoing discussion regarding the role of the spin-orbit coupling (SOC). For example, in Ref.12 the experimental work concentrated on characterizing the electronic structure outside the nodal planes depicted in Fig.1, where the altermagnetic spin-splitting can be ignored. On one hand, the authors of this manuscript claim that SOC is weak for Cr 3d electrons and plays a minor role in the low-energy band structure. On the other hand, the theoretical calculations in Fig. 2g shows that for the "AM with SOC" case

the spin polarization appears completely washed out compared to Fig.2k, which is the “AM without SOC”. The fact that SOC is not needed does not mean that it is not present. What is the message here: does the SOC destroy the spin polarization? In addition, the discussion related to Fig. 2j “(NM) with SOC” is missing in the main text. It is unclear how is the non-magnetic (NM) case implemented and how does this connect to panels j and k in Fig.2?

We thank the reviewer for pointing out this error in Fig. 2g – we made a mistake in our script on the calculation of $\langle S_z \rangle$, which led to incorrect $\langle S_z \rangle$ values shown in the old Fig. 2g. We are very sorry for this mistake and confusion that it caused. We have now corrected this error and updated the figures accordingly (which includes Fig. 2, Fig. 3 in the manuscript, Fig. S6, Fig. S7 and Fig. S9 in SI). Comparing Fig. 2g and Fig. 2k, it is clear now that the spin polarization near E_F for the “AM with SOC” case is indeed very close to that of “AM without SOC”, except at special momentum/energy positions (see more discussions below). This proves that the SOC in CrSb is indeed very small and does not affect the band dispersion or spin polarization near E_F significantly. We are very grateful to the reviewer for pointing out this important issue, which otherwise could cause confusion for the readers.

To understand the weak SOC effect on the spin polarization, we have further calculated the spin polarizations along x, y and z directions, i.e., $\langle S_x \rangle$, $\langle S_y \rangle$ and $\langle S_z \rangle$, and the results are added as the new Fig. S12 in SI (the relevant discussions are included in the figure caption). Specifically, there are two noticeable effects on the spin polarizations from SOC:

1. Away from the nodal planes and far below the Fermi level, e.g., at $k_z = 0.28 c^*$ and $E < -1.5$ eV (see Figs. S12a-c), the inclusion of SOC makes $\langle S_z \rangle$ deviate from the quantized value of $\pm 1/2$, and at the same time $\langle S_x \rangle$ and $\langle S_y \rangle$ become non-zero. According to our DFT calculations, these bands far below the Fermi level contain appreciable contributions from Sb p orbitals and that is why obvious SOC effects are present. We mention that in these energy/momentum regions, the in-plane spin components ($\langle S_x \rangle$ and $\langle S_y \rangle$) are still much smaller than that of $\langle S_z \rangle$ (please note the different color scales in Figs. S12a&b and Fig. S12c). We also emphasize that the spin polarization near E_F is much less affected by SOC. We have revised the manuscript to illustrate this point (page 4, middle): “... However, far below E_F where appreciable contributions from Sb 5p orbitals are present, e.g., $E \sim -2$ eV, the inclusion of SOC makes $\langle S_z \rangle$ deviate from $\pm 1/2$ appreciably and leads to noticeable in-plane spin polarization (see Fig. S12 in [63]). ...”
2. Introducing SOC breaks the spin degeneracy in specific regions of original nodal planes, resulting in a small gap opening. In the case of CrSb, only k_z axis remains spin-degenerate when considering SOC. For the M-G-M path at $k_z = 0$ plane, the SOC induces a very tiny splitting compared to the case without SOC (see Fig. S7). While the z-component spin polarization ($\langle S_z \rangle$) remains zero, there is a small in-plane spin polarization due to SOC (see Figs. S12d-f). This is similar to the weak antiferromagnetic splitting discussed in Ref. [13]. However, such splitting is too weak to observe experimentally. We have added the following texts to explain this point (page 4, middle): “Indeed, if the SOC in CrSb were strong, it would transform the aforementioned nodal planes into a nodal line along the Γ -A direction [40].”, “... Another effect from SOC is the tiny band splitting at $k_z = 0$ and concomitant non-zero in-plane spin components $\langle S_x \rangle$ and $\langle S_y \rangle$ (see Fig. S7 and Fig. S12 in [63]), which is called weak antiferromagnetism in [13]. Such small splitting cannot be resolved in our experiments (comparing Figs. 2e,h).”

Finally, the purpose of Fig. 2j is to illustrate the importance of altermagnetic order for explaining the observed band dispersion. Specifically, the “AM with SOC” and “AM without SOC” calculations in Fig. 2g/k are in good agreements with the experimental results in Fig. 2a/d, but the “NM with SOC” calculation in Fig. 2j obviously deviates from experiments. Therefore, such a systematic comparison between experiments and calculations provides direct evidence for the altermagnetic order from the study of electronic structure. To further clarify this point, **we have added description on fig. 2j in the manuscript (page 4, second paragraph): “... The clear difference between Fig. 2g and Fig. 2j, when comparing with experimental spectra in Fig. 2d, provides direct evidence for the altermagnetic order from the band dispersion.”**

B) Unfortunately, the most important Ref.41 is only marginally mentioned in the manuscript. The authors in Ref.41 (presumably) intentionally concentrated on the P-Q-P rather than M-G-M directions in order to provide the experimental evidence for AM splitting. Concerning the point A, did the authors consider to calculate the “AM with SOC” to check the spin degeneracy predicted in their TB-model along the P-Q-P directions? Such a comparison would be extremely useful to elucidate the role of the AM splitting in CrSb 3D electronic band structure.

We thank the reviewer for the suggestion. To facilitate direct comparison between our ARPES results and those in old ref. [41] (now ref. [51]), we have marked **the P-Q-P direction (defined in Ref. [51]) in our Figs. 2l-n**. In fact, the P-Q-P direction in Ref. [51] was accessed by in-plane angular scan under an optimally chosen photon energy, while the same momentum direction was probed by photon-energy-dependent scan in our case. Therefore, the altermagnetic splitting presented in Figs. 2l,m is essentially along the same momentum path as Fig. 3 & Fig. 4 in Ref. [51]. In Fig. 2n, the altermagnetic band splitting from the “AM with SOC” calculation was plotted, including the expected spin polarization. The good agreement between experiments (Figs. 2l-m) and calculations (Fig. 2n) provides direct evidence for the altermagnetic splitting in CrSb, which is overall consistent with Ref. [51]. To include a detailed comparison with Ref. [51] in our paper, **we have added more discussions in the manuscript**, which now reads: **“... Along the P-Q momentum direction as defined in [51] (also labelled in Figs. 2l-n), the raw data already show a double-string feature (see arrow in Fig. 2l) with closed ends at both P and Q points, which can be better identified from the second derivative in Fig. 2m. ... Our observed splitting along the P-Q direction is overall consistent with the previous ARPES study from (100)-oriented CrSb films [51], although the measurement geometries are different.”**

By the way, we would like to emphasize that thanks to the higher counts and resolution from ultraviolet light ARPES, our measurements allow us to reveal the altermagnetic splitting in 3D momentum space (Fig. 2 and Fig. 3), which was not covered in Ref. [51].

C) Within the TB-approximation, loosely speaking, the atoms are free, whereas the electrons are tightly bounded. The fact that the AM nearest-neighbor effects in spin-splitting arises from strong third-nearest-neighbor hopping mediated by Sb ions is an interesting aspect of the

manuscript. This message, however, comes out under significant computational machinery: starting from DFT (VASP), across PBE parametrization of GGA, to finally reach out to a TB-model Hamiltonian with maximally projected Wannier function method. The authors claim that their results “show that altermagnetism plays a dominant role in the observed band structure, while the SOC effect is not very important”. The main problem I have here is that based on my critique in point A, the 1 eV spin-splitting in the M-G-M direction in Fig. 2g, is simply zero or close to zero. How is the 3NN AM splitting connected with the vanishing spin splitting indicated by green arrow in Fig.2g?

We thank the reviewer for asking this question. As we have discussed in our answer to point A, we mistakenly plotted the wrong $\langle S_z \rangle$ values in our initial manuscript. In the revised version, the spin polarization along the z axis ($\langle S_z \rangle$) is now much closer to the expected value of $\pm 1/2$ (see **new Fig. 2g**), particularly for the bands near E_F . Therefore, this justifies our point that the SOC effect in CrSb is not very strong, and hence a TB model without considering SOC can provide a good explanation for the observed altermagnetic splitting near E_F .

Based on our TB model analysis (details are included in the supplementary file), we further uncovered the important role of the 3NN hopping mediated by the Sb ions. We are glad that the reviewer appreciates our theoretical approach in this paper.

Furthermore, there are following minor points related with the manuscript:

a) The ARPES data presented are in the UV-range, hence they are surface sensitive. Despite this, the 3D band maps at selected photon energies in Fig.2a-f faithfully reflect the bulk-sensitive DFT calculations. However, this is not the case for the AM-splitting discussed in Fig.2l-m: it is impossible to see the spin splitting in the indicated k_z -cut. Data summarized in Fig.3, however, do support this splitting. I would suggest mentioning this in text. As these data are related to CrSb g-wave altermagnetism, which is very difficult to observe in Fig.3a, I also suggest the authors to refer to recent work in <https://arxiv.org/abs/2405.12575>, where the 6-fold symmetry in the constant energy map is indeed well observed.

We thank the reviewer for the comments and questions. Below we answer them point by point:

1. The issue with altermagnetic splitting in the $k_z - k_x$ map in Figs. 2l-m. Our original discussion related to Figs. 2l-m might not be very clear, which could cause confusion. Here Fig. 2l is the constant-energy cut in the $k_z - k_x$ plane (different from Figs. 2a-f), and it is extracted from ARPES data taken with different photon energies (see Fig. S8 in SI). Therefore, the altermagnetic splitting here is manifested through momentum-dependent band splitting (or the double-string feature along P-Q) as marked by arrow in Fig. 2n: the separation between spin-up (red) and spin-down (blue) bands is zero at $k_z = 0.5c^*$ (L-A-L), becomes maximized near $k_z \sim 0.25 c^*$ and eventually gets close to zero at $k_z = 0$ (M- Γ -M). Such splitting can already be identified in the raw data in Fig. 2l (see arrow), although we agree with the reviewer that the splitting can be obscured by the background. To enhance the visibility of the altermagnetic splitting, **we have now included a second derivative of the $k_z - k_x$ map in Fig. 2m, where the altermagnetic splitting can be better identified**. The experimental result is also well captured by the DFT calculation shown in Fig. 2n. We have also added the following texts to explain the altermagnetic splitting in the $k_z - k_x$ map: **"... its second derivative and the calculated contour from DFT are displayed**

in Figs. 2m and 2n, respectively. ... Along the P–Q momentum direction as defined in [51] (also labelled in Figs. 2l–n), the raw data already show a double-string feature (see arrow in Fig. 2l) with closed ends at both P and Q points, which can be better identified from the second derivative in Fig. 2m. ..."

2. The issue with altermagnetic splitting in the $k_x - k_y$ cut in the old Fig. 3a. Due to the weak photoemission cross section near E_F under 80 eV photons, the $k_x - k_y$ map in the old Fig. 3a is indeed not very clear. To improve the readability of our work, we have now included a new FS map taken with 105 eV photons as the **new Fig. 3b (the old Fig. 3b now becomes Fig. 3a)**, which is close to the k_z value as 80 eV photons. The six-fold symmetry and flower-like Fermi contours can now be much better resolved. More texts are added to discuss this map (see also reply to question 2 from the second reviewer).
3. By the way, the link <https://arxiv.org/abs/2405.12575> is actually our paper. We guess that the reviewer was referring to other ARPES works that were posted on arXiv around the same time or slightly later (arXiv:2405.12679; arXiv:2405.12687; arXiv:2405.14777). Our updated FS map in Fig. 3b is of similar quality as those papers. **We have also added a note at the end of the manuscript to mention these ARPES works: "Note: during the review stage of this paper, we became aware of the other ARPES works on CrSb posted on arXiv [66–68], which also support the altermagnetism in CrSb."**

b) In Fig.2, panels a,d,g,j,k I would suggest not using the $\bar{G} - \bar{M}$ notation, just like in panels b,e,h; because these ARPES data and calculations are at specific k_z values in the 3D band electronic band structure.

We thank the reviewer for this suggestion. We understand that it would be best to use the notation defined in three-dimensional Brillouin zone, in order to avoid possible confusion. However, since the momentum points where the altermagnetic splitting is maximal are not high-symmetry points, there are no commonly used notations for these momentum points. In addition, due to the well-known k_z issue in ARPES measurements, it is quite common in the ARPES literatures to use the notation of surface Brillouin zone, e.g., $\bar{\Gamma}\bar{M}$ with k_z values, to specify the momentum direction of ARPES data. Therefore, we decide to keep the notation of surface Brillouin zone, but we have also made the following clarifications based on the reviewer's suggestions:

1. We have added **the definition and notation of surface Brillouin zone in Fig. 1e** to clarify the definition of surface Brillouin zone: "... The surface BZ for (001)-oriented CrSb, which is the projection of three-dimensional BZ onto the (001) surface, is labelled in (e) with the high-symmetry points $\bar{\Gamma}$, \bar{M} and \bar{K}"
2. We have **added the corresponding k_z values in all figures**, to unambiguously specify the momentum points in three-dimensional Brillouin zone.

c) In Fig.1d, I do not know how to interpret the $k_z = nc^*/2$ label

We are sorry for this confusion. Here c^* is the reciprocal lattice constant of CrSb along the c axis ($c^* = 2\pi/c$) and n is an integer. Therefore $k_z = nc^*/2$ is used to label nodal planes perpendicular with the c axis, i.e., the light-blue horizontal planes in Fig. 1d, where the altermagnetic splitting vanishes

(ignoring SOC). To clarify this issue, we have added the following texts in the figure 1 caption: “The $k_z = nc^*/2$ nodal planes, where $c^* = 2\pi/c$ and n is an integer, are protected by $[s_2^C || M_z]$ combining with translation symmetry.”

In conclusion, although the presented manuscript is a valuable contribution to CrSb altermagnetism, I am not convinced that it is suitable for a broader audience and thus for publication in Nature Communications in the present form. With adequate response to my critique A,B,C, however, I'd like to reconsider my recommendation.

We thank the reviewer for providing comments and criticisms that help us improve our paper, particularly on the issue of spin polarization and SOC. We believe that these issues are now resolved and the paper is much improved.

Reviewer #2 (Remarks to the Author):

The authors have studied one of the altermagnet candidates, CrSb, using synchrotron-based ARPES. Compared to previous experimental work on CrSb thin films [Ref.41], this study resolves spin-splitting more clearly, with a maximum separation of 1 eV. They also revealed g-type spin polarization by comparing their results with ab initio calculations. I believe this work merits publication in Nature Communications, provided the following concerns are addressed. All of my concerns stem from the poor resolution in some of the experimental results, which currently hinders a clear conclusion regarding the match between theoretical and experimental findings.

We thank the reviewer for appreciating the importance of our paper. Following the reviewer's suggestions, we have added additional experimental data and analysis to better illustrate the altermagnetic splittings. We shall discuss these results in details below.

1. Although the split feature is recognizable in Fig. 2a & d ($h\nu=80$ eV), its visibility in the constant energy contour is very poor. Despite the authors' optimistic statement that "The experimental results and DFT calculations agree quite well" (p.4), the string-like feature with closed points at both ends is hardly visible. I suggest creating a second derivative image to improve visibility

We thank the reviewer for this suggestion. We have followed the reviewer's advice and created a second derivative of the image plot of Fig. 2l, which is now displayed as **the new Fig. 2m**. Now the double-string feature (marked by **a white arrow in Fig. 2m**) with closed points at both ends can be better identified, which is well captured by the DFT calculation in Fig. 2n. **The figure caption and relevant discussions are updated accordingly**, which now reads: "... its second derivative and the calculated contour from DFT are displayed in Figs. 2m and 2n, respectively. ... Along the P-Q momentum direction as defined in [51] (also labelled in Figs. 2l-n), the raw data already show a

double-string feature (see arrow in Fig. 2l) with closed ends at both P and Q points, which can be better identified from the second derivative in Fig. 2m. ..."

We mention that the k_z dispersion of the bulk bands is often complicated by the large variation of photoemission cross section with photon energy, which can obscure the spectral features in the $k_x - k_z$ map. This is perhaps why the raw data in Fig. 2l is not very clear, but the second derivative can sharpen the bands and allow for better visualization of the splittings. We are very grateful to the reviewer for his/her suggestions.

2. The same issue occurs in the constant energy surfaces in Fig. 3a & b. The pair of deformed triangles (dashed lines) seen in the calculations cannot be observed in the experimental results. This part is crucial for supporting the claim of g-wave type spin texture, but it is currently unconvincing.

We thank the reviewer for this comment. Since the photoemission intensity under 80 eV photons is very weak near E_F , the $k_x - k_y$ constant-energy contours in the old Fig. 3a & b are not well resolved. We have performed additional experiments in a wider photon energy range to search for better Fermi surface maps. Our new Fermi surface map taken with 105 eV photons, corresponding to $k_z \sim 0.2 c^*$, are now included in Fig. 3b (the old Fig. 3b now becomes Fig. 3a). Now the pair of deformed triangles at E_F can be better visualized from the raw data, providing direct evidence for the large g-wave antiferromagnetic splitting at E_F . The following texts are also added to discuss these points: "... Since the photoemission intensity near E_F is very low under 80 eV photons, we further show the Fermi surface map taken with 105 eV photons in Fig. 3b, which corresponds to $k_z \sim 0.2c^*$. Here the six-fold flower-like pockets expected from DFT calculations are clearly visible from the raw data, highlighting the large antiferromagnetic splitting right at E_F"

To address the poor resolution in the constant energy contours, it might be worthwhile to perform spin-resolved ARPES to confirm the type of spin-polarization. However, this could be challenging due to the mixture of different magnetic domains. A previous experiment on MnTe attempted this but failed to resolve it well, resulting in only a tiny spin polarization [Ref.12]. As the authors claim in this manuscript (p.6) "Firstly, the 3d spins in CrSb are aligned along the c axis, in contrast to the in-plane 3d spins in MnTe. This eliminates multiple band splittings caused by the different in-plane domains as reported in MnTe [12,14,52] and allows for clear identification of antiferromagnetic splitting in CrSb.", clearer spin polarizations are highly expected for CrSb.

I wonder if the authors have already attempted such measurements. If so, I recommend including some comments on the spin polarization measurements.

We thank the reviewer for this suggestion. To provide additional evidence for the spin splitting, we have recently performed spin-resolved ARPES measurements and have obtained direct evidence of spin polarization. Therefore, we add a new section ("Evidence of spin polarization from spin-resolved ARPES") and a new figure (Fig. 4) in the manuscript to discuss these results. The Method section is updated accordingly. As shown in the new figure 4, there is a clear spin polarization signal associated with the antiferromagnetic splitting and the polarization direction is consistent with

theoretical predictions. This provides direct support for the altermagnetic splitting in CrSb. Nevertheless, the magnitude of spin polarization is small and it is likely caused by mixed altermagnetic domains or large momentum broadening in the ARPES spectra. All the detailed discussions are included in this new section in the manuscript.

Reviewer #3 (Remarks to the Author):

The article provides a three-dimensional k-space mapping of the CrSb. However, I have some criticism that should be addressed by the authors before publication: As a summary, the article contains valuable information about these CrSb. Especially remains challenging in the three-dimensional k-space mapping. Their interesting results are well captured between experiments and theories. My opinion of the paper is quite positive, but I would like some moderate revisions before acceptance.

We thank the reviewer for appreciating the importance of our work. We will answer the reviewer's questions point by point below.

Below, you find my questions/remarks:

1, In ab initio calculations, the antiferromagnetic splitting of CrSb is maximized in the in-plane $\Gamma - M$ direction with $k_z \sim 0.25c^*$, what is the reason for this result? And why this trend?

We thank the reviewer for asking this question. Due to the symmetry requirement, the altermagnetic splitting along the in-plane $\bar{\Gamma}\bar{M}$ direction is forced to be zero at $k_z = 0$ and $k_z = 0.5c^*$ (ignoring SOC). In addition, the crystal structure of CrSb exhibits equally spaced alternating Cr and Sb layers. Therefore, it is natural to expect that the k_z position where the altermagnetic splitting is maximized is close to $0.25c^*$, i.e., the middle of two nodal planes.

Nevertheless, the exact location of maximal altermagnetic splitting is dependent on the crystal structure and the orbital characters. Therefore, it can be different from the exact middle point between nodal planes. To clarify this point, we have added the following texts: "... The k_z position where the altermagnetic splitting is maximized is close to the midpoint between two nodal planes at $k_z = 0$ and $0.5c^*$, which is likely related to the detailed crystal structure of CrSb, although such a coincidence is not enforced by any specific symmetry."

2, In Fig2a-c, the mechanism of valence band changes at three different photon energies should be explained?

We thank the reviewer for this suggestion. To clarify this point, we have added the following texts to explain the observed valence band changes in Figs. 2a-c: "... Based on the standard understanding of photoemission [61, 62], ARPES measurements under different photon energies probe electronic states with different k_z values. The conversion between photon energy and k_z is

determined by the inner potential V_0 , which is the difference between the crystal potential and the vacuum level. ... These changes in band dispersion with photon energies can be attributed to different k_z 's, as discussed above. In addition, the photoexcitation probability for the probed valence bands can also vary with the photon energy and the electron energy/momentum, leading to different photoemission intensity. ..."

3, In Fig3a-b, could similar flower-like hole-type pockets be observed at photon energies other than 80 eV photons? I doubt it.

We thank the reviewer for asking this question. In principle, the flower-like hole-type pockets should be observable in other photon energies too, as long as the photon energy probes the k_z range where the altermagnetic splitting is large enough for experimental observation. In reality, due to the experimental broadening (such as k_z broadening, uneven cleaved surface, etc.), such flower-like hole-type pockets might not be easily identified. In the revised manuscript, we **have added a new Fermi surface map taken with 105 eV photons in Fig. 3b (the old Fig. 3b now becomes Fig. 3a)**, where the flower-like hole-type pockets at E_F can be better visualized. Texts are also added to discuss the new data: "... Since the photoemission intensity near E_F is very low under 80 eV photons, we further show the Fermi surface map taken with 105 eV photons in Fig. 3b, which corresponds to $k_z \sim 0.2c^*$. Here the six-fold flower-like pockets expected from DFT calculations are clearly visible from the raw data, highlighting the large altermagnetic splitting right at E_F"

4, At $q_2 + q_3$ greater than about 0.2 eV, the effects of q_1 and q_4 become significant; what is the physical mechanism for this transition? In addition, what other factors besides the 3NN hopping may have an effect on the altermagnetic splitting?

We thank the reviewer for these questions. The reason why the effects of q_1 and q_4 become significant at larger $q_2 + q_3$ is related to the multiorbital nature of CrSb. Specifically, two sets of Cr 3d orbitals are needed (ϕ_1 and ϕ_2 in our tight-binding model) to describe the conduction bands near E_F in CrSb, and both of them exhibit altermagnetic spin splitting. When $q_2 + q_3$, which directly controls the altermagnetic splitting within one set of band (defined as ΔE_1), is large, the "effective" spin splitting near E_F at a specific momentum position (defined as ΔE_2) can be affected by nearby bands with similar spin splittings. To demonstrate this point more clearly, we have made the following changes to the paper:

1. In the **new Fig. 5c-d**, we have added two calculations of the tight-binding bands with different $q_2 + q_3$ and $q_1 + q_4$ values to directly illustrate their effects on the band dispersion. It is now clear from the plots that $q_2 + q_3$ dictates the altermagnetic splitting within one band, while $q_1 + q_4$ controls the overall bandwidth. We also make two distinct definitions of spin splitting: the altermagnetic splitting within one band (ΔE_1) and the splitting of opposite-spin bands closest to E_F (ΔE_2). While ΔE_{1U} has a clear meaning in the physics of altermagnetism, ΔE_{2U} is directly relevant to the spintronic applications.
2. In the **new Fig. 5e-f**, we have added the calculations of both ΔE_{1U} and ΔE_{2U} as a function of $q_2 + q_3$ and $q_1 + q_4$. Now it is clear from these plots that ΔE_{1U} is approximately proportional to $q_2 + q_3$, while ΔE_{2U} becomes dependent on $q_1 + q_4$ when $q_2 + q_3$ becomes larger. This is

because when $q_2 + q_3$ becomes large, the top conduction bands with spin splitting will now cross the Fermi level and reduce ΔE_{2U} .

3. In the **new Fig. 5g-h**, we have also included the calculations of ΔE_{1U} and ΔE_{2U} along the $\overline{\Gamma M}$ path ($k_z = 0.28 c^*$) with different q values (while fixing the ratio of $q_1:q_2+q_3:q_4$). These plots reveal the monotonic increase of spin splittings under uniform compression.
4. **The discussions relevant to Fig. 5 are also revised accordingly.**
5. We have **added a new figure in SI (new Fig. S13)**, to illustrate the detailed change of band dispersion with different $q_2 + q_3$ and $q_1 + q_4$ values. This figure, together with its figure caption, can help clarify the role of $q_2 + q_3$ and $q_1 + q_4$ on the spin splittings near E_F .

While the 3NN hopping plays a dominant role in the altermagnetic splitting, there are other factors that can also contribute to or modify the altermagnetic splitting. Firstly, as discussed previously, the SOC can affect the altermagnetic splitting, leading to additional (small) band splittings and reduction of spin polarizations. Secondly, the higher-order hopping process beyond the 3NN hopping can also give rise to small altermagnetic splitting, although its magnitude should be much less than the 3NN hopping.

5, What is the matrix representation of symmetric operations?

We thank the reviewer for the question. **We have listed all the matrix representation in our defined basis in SI section 3: Matrix representation of symmetry operation.**

6, There is a clear $\{C_z_{\infty} || E\}$ symmetry operation regarding spin in the structure, which is the infinitesimal rotation of spin about the z-axis that the author did not consider.

We thank the reviewer for the question. We did not write it explicitly in our article. This infinitesimal rotation $\{C_z_{\infty} || E\}$ can protect spin as a good quantum number with a momentum-independent spin-quantization axis (here is z axis) across the whole Brillouin zone. It is just like that we add a phase in the TB model, it will not change the eigenvalues of TB models.

7, The format of the references is extremely poor, with many of them having inconsistent formats and missing some theoretical predictions, especially the recently published such as APL, et.al

We thank the reviewer for this criticism. We have added quite a few recent and related theoretical papers as references, including **Refs. [7, 40, 41, 47, 49]**. The texts are revised accordingly. We have also checked the format of all our references and corrected some mistakes.

Re: NCOMMS-24-30428A

Authors: Guowei Yang et al.

Dear reviewers,

Thank you very much for your constructive comments and recommendations to accept our manuscript. Following your suggestions, the manuscript and SI are revised accordingly. Below we addressed the reviewers' questions in a point-by-point fashion (All the revised parts in the manuscript are marked in red).

Reviewer #1 (Remarks to the Author):

I am pleased to say that the authors reacted to my critique, answering most of the questions I raised during the first review. With the additional SARPES data, the authors provided valuable information in understanding the altermagnetic (AM) spin splitting in CrSb. Based on the comments and critiques reacted/answered also to other reviewers, my opinion is that the manuscript is suitable for publication in Nature Communications. However, I take the liberty to address a few important revisions before acceptance.

We thank the reviewer for appreciating the importance of our work and recommending its publication in Nature Communications.

Let me address the manuscript from a broader perspective first. The manuscript is, to a large extent, difficult to read. Fair enough, it combines spin- and angle-resolved photoemission (SARPES) mapping in 3D momentum space, which for AM systems is the experimental method of choice, with TB ab-initio calculations. As I mentioned during the first review round, these calculations give insight into specific hopping terms that show the relevance of 3NN terms. However, this part of the manuscript appears disconnected from the SARPES message. The problem is that to underpin this message in text with different MnTe and VSb₃S₆ systems, the outcome becomes rather speculative. For example, contrary to CrSb, in MnTe the AM spin-splitting at Fermi level (EF) is based on Te p-orbital states because the Mn d-states are deep in the valence band. VSb₃S₆ is a vdW 2D-like system, hence again significantly different in terms of orbital ordering. The systems indeed share the *g*-wave AM order parameter outside the nodal planes, but the role of the orbital interplay in the AM spin-splitting magnitude close to EF would need detailed TB-analysis of these systems, which is not the scope of the manuscript.

We agree with the review that our TB model cannot directly apply to all kinds of altermagnets, since the different nature of magnetism and multiple bands in specific materials can complicate the altermagnetic splittings in real systems. Therefore, we have made the following changes to the manuscript following the reviewer's suggestions:

- (Page 8, end of first paragraph) By contrast, the observed altermagnetic splitting in MnTe is ~ 0.5 eV and is located far below E_F , which is mainly derived from Te p orbitals [12,13]. We note that our TB model might not be directly applicable to MnTe, due to different orbital character and nature of magnetism.
- (Page 8, second paragraph) We delete the following statement: ~~The large difference of the splitting magnitude in CrSb and VNb_3S_6 highlights the important role of interlayer electron hopping (the 3NN hopping in this case), which is mediated by coupling with the nonmagnetic ions, as discussed above.~~
- (Page 8, second paragraph) We add the following sentence to clarify: **Since the magnetic moments of V are antiparallel between two neighboring layers, the altermagnetic splitting in VNb_3S_6 also originates from the interlayer hopping.** Yet its much larger lattice constant $c = 12.17$ angstrom [65] and weak interlayer coupling between V and NbS_2 layers result in a very small splitting of a few tens of meV (see Fig. S11 in [63]).

On the other hand, the unusual NN, NNN, and 3NN hopping amplitudes in CrSb stem for a dichotomy in “localized” and “delocalized” magnetism in AM systems, because the 3NN hopping appears to be mediated by Sb orbital states. Let me emphasize that, in contrast to semiconducting MnTe, CrSb is a metal, hence one would a priori expect an *itinerant* type of magnetism mediated with conduction band. In that sense, the TB-analysis indicates that CrSb becomes an *itinerant* AM in a very specific way. Did the authors consider the $q_2 + q_3$ terms arising from the direct and assistant hopping processes to be related to such a dual AM character?

We thank the reviewer for raising this interesting question. Indeed, our experiments and calculations indicate that the magnetism in CrSb is mainly of itinerant type. On the other hand, magnetism in correlated 3d-electron systems often shows dual characters with both localized and itinerant components. While the direct and assistant parts of q_2+q_3 terms have been calculated and listed in the SI (table 4), their deep connection with the localized/itinerant character of magnetism remains to be studied. We believe that this topic will be very interesting for future studies.

Next, I address the SARPES part of the manuscript. The 3D k_z mapping summarized in Fig.2 is convincing, the panel m now reflects the peculiar k_z -dispersion quite well. The authors refer to AM spin-splitting for $k_z=0.28c^*$, where the “strong” AM splitting is expected at 80 eV photon energy. The relevant ARPES band maps are in Fig.2a and Fig.4a, which should be equivalent. However, it is evident that the data in Fig.2a is of better quality compared to Fig.4a. I understand this is because data in Fig.4a were measured from a different sample prepared for the SARPES experiment. If so, this should be mentioned in the text because the authors qualitatively managed to reproduce comparable ARPES band maps to further examine the CrSb AM spin-splitting.

We thank the reviewer for this suggestion. Indeed, the experimental results in Fig. 2 and Fig. 4 come from different samples measured in different machines, but they show consistent results. The different quality of spin-integrated ARPES data can be attributed to different sizes of beam spots and different measurement setups (see Methods). We have changed our manuscript accordingly:

- (Page 5, third paragraph): To verify the spin polarization of the antiferromagnetically split bands, **separate spin-resolved ARPES measurements**... **The different data quality in Fig. 4a and Fig. 2a can be attributed to different experimental conditions.**

Next, the arrows in the second derivative map in Fig.4b indicate the CrSb AM double-peak structure. If the “double-peak” structure refers to the red/blue arrows; and not the broad spectral intensity for $k_{\parallel} < 0$ and $k_{\parallel} > 0$, respectively, then the statement:

It is clear that both spin-up and spin-down MDCs exhibit a double-peak structure with large broadening, corresponding to the spin-up and spin-down bands labelled in Fig. 4b.

...is incorrect, because the broad antisymmetric MDC spectral weight does not resolve the double-peaks. Instead, the S_z up/down signal in Fig.4c features asymmetric spectral distribution for $k_{\parallel} < 0$ and $k_{\parallel} > 0$, respectively. The two bands for $k_{\parallel} < 0$ and $k_{\parallel} > 0$ are not resolved as in Fig.2a, and neither in the SARPES data in Fig.4d. Importantly, data in Fig.4c,d are missing the error margins. They would confirm that the double-peak structure can hardly be resolved in SARPES. Yet, the MDC has the expected spin-polarization asymmetry for $k_z = 0.28c^*$ for $k_{\parallel} < 0$ and $k_{\parallel} > 0$. This is important because, within the MGM mirror plane, this MDC in AM spin-splitting must change sign. The ultimate proof of the CrSb AM spin-splitting would be to measure the same MDC for opposite $k_z = -0.28c^*$, which must have opposite S_z in MDC distribution. Did the authors manage to measure this dataset with different photon energy?

We thank the reviewer for the comments and questions. Below we answer them point by point:

- For the statement “*It is clear that both spin-up and spin-down MDCs exhibit a double-peak structure with large broadening, corresponding to the spin-up and spin-down bands labelled in Fig. 4b*”: here the “double-peak” structure refers to the broad spectral peaks at $k_{\parallel} < 0$ and $k_{\parallel} > 0$, respectively, NOT the separate blue/red arrows. So this statement is valid.
- Regarding the antiferromagnetic splitting: the splitting is already clear in the raw data of Fig. 2 - see Fig. 2a and Fig. 2l. In Fig. 4a, the antiferromagnetic splitting is not obvious in the raw spin-integrated data, due to the large spectral broadening as discussed above. However, the splitting can be identified in the second derivative (Fig. 4b), similar to Fig. 2. In spin-resolved measurements, the antiferromagnetic spin splitting is manifested by the shifting of the center of the broad peaks at both $k_{\parallel} < 0$ and $k_{\parallel} > 0$ (Fig. 4c), corresponding to the switching from red to blue arrows in Fig. 4b.
- Following the reviewer’s suggestion, **we add error bars to Fig. 4d**. The caption is changed accordingly: ... **Error bars are defined as the standard error of polarization at each k point between repeated scans.**
- Additional SARPES data: due to the limited beamtime and long counting time for SARPES, we have not tried different photon energies to verify the k_z dependence of spin polarization. This will be our future task.

Minor issues:

- I would also recommend adjusting the cartoon figure in Fig.1d: the red and blue shades intuitively connect with the S_z spin polarization in the nodal planes discussed in the text, suggesting a symmetric spin polarization outside the $k_z = 0$ nodal plane. This is misleading. The symmetric AM

spin splitting can occur only in the diagonal direction -LGL. If the authors did not intend to label the AM spin polarization of the four nodal planes in the cartoon figure, I would recommend using different color shades.

We thank the reviewer for this suggestion. To prevent misinterpretations, we **changed the color of nodal planes in Figure 1d** (and **arrows in Figure 1c**), accordingly.

- *The two opposite-spin sublattices in altermagnets should be connected by a (screw) n-fold rotation (C_n): n cannot be an arbitrary integer, for CrSb it is 6*

We have changed the manuscript accordingly (Page 3, first paragraph): The two opposite-spin sublattices in altermagnets should be connected by a (screw) n-fold rotation (C_n , **n depending on lattice/spin symmetries**) or (glide) mirror (M) symmetry in real space...

- *When SOC comes into play, the $SU(2)$ symmetry is broken and the bands with opposite spins will couple to each other. I do not understand the meaning of the $SU(2)$ symmetry, it is not mentioned nor explained in the text.*

To clarify, we have changed this sentence: When SOC comes into play, the $SU(2)$ symmetry is broken and the bands with opposite spins will couple to each other, **making S_z not a good quantum number**.

- *Indeed, if the SOC in CrSb were strong, it would transform the aforementioned nodal planes into a nodal line along the $\Gamma - A$ direction [40]. CrSb possesses topological Weyl nodal structures which do not necessarily require strong SOC according to Ref.68. Therefore, I recommend dropping this statement.*

We thank the reviewer for this suggestion. Strictly speaking, nodal planes transform into nodal line once the SOC is nonzero. However, if SOC is weak, such effect cannot be observed experimentally, and ARPES measurements still yield nodal planes, as demonstrated in our work. To clarify, we have changed our statement accordingly:

- Indeed, if the SOC in CrSb were strong, it would transform the aforementioned nodal planes into **an experimentally observable** nodal line along the $\Gamma - A$ direction [40].

In conclusion, the manuscript contains valuable findings in understanding the CrSb altermagnetism. I recommend the publication for Nature Communications in an improved version of the manuscript along the lines mentioned above. Also, the manuscript is difficult to read for a broad audience. In that sense, I would recommend revising the paragraph on the *Mechanism behind the large altermagnetic splitting*. It is advisable to reconsider the interpretation of the TB model of CrSb because the way this message extrapolates to MnTe and VSb3S6 is speculative. If the authors have more ARPES data supporting the ultimate AM splitting in 3D CrSb mentioned above, I would recommend using these data in this manuscript.

We thank the reviewer for the comments and questions. Below we answer them point by point:

1. The section on *Mechanism behind the large altermagnetic splitting*: This section highlights one of the novel aspects of our work and aims to provide an in-depth understanding of altermagnetic splittings in a concise manner, which can be interesting to the broad audiences working in the field of altermagnetism. All the details of calculations and TB modellings are included in the SI. We believe that our current writeup is already a good balance between key scientific discoveries, broad interests and technical details.
2. The discussion on the differences between CrSb, MnTe and VNb₃S₆ has been now revised (see reply above), according to the reviewer's suggestions.
3. Additional SARPES data: due to the limited beamtime and long counting time for SARPES, we have not tried different photon energies to verify the k_z dependence of spin polarization. However, we believe that we have already presented extensive sets of experimental data, which provide compelling evidences of large altermagnetic spin splittings in CrSb.

Reviewer #2 (Remarks to the Author):

I would like to express my deep appreciation for the authors' tremendous efforts in conducting another experiment to detect the spin polarization of the split energy bands, which appears to align well with theoretical expectations. Although the magnitude of the spin polarization is smaller than the predicted value, it is reasonable and sufficient to support their claim of observing spin-split altermagnetic bands. The newly added Fermi surface contour measured at 105 eV is commendable as well. I believe they have done their utmost to detect spin polarization.

We thank the reviewer for appreciating our experimental efforts.

I understand that such photoemission experiments cannot be performed under strong external magnetic fields, in contrast to magnetic circular dichroism experiments. Additionally, obtaining a single magnetic domain for antiferromagnetic materials poses challenges, as spin-resolved photoemission requires, in principle, a single "remanent" magnetic domain.

As the reviewer pointed out, the challenge of SARPES is to obtain single-domain altermagnet. There are a few proposals for achieving single-domain altermagnet, but they are not easily compatible with standard ARPES measurements. At present, we rely on synchrotron-based SARPES with a small beam spot, to selectively probe an area with preferably one domain. Performing SARPES measurements on pure single-domain altermagnet will be an important topic for future studies.

I find that the authors have addressed all concerns raised by the three referees, including myself, in a reasonable manner. Therefore, I recommend this work for publication in Nature Communications. We thank the reviewer for the positive feedback and the recommendation for publication of our manuscript.

Reviewer #3 (Remarks to the Author):

The author has responded to each of the questions I raised, and I think they can now be accepted by NC.

We thank the reviewer for the positive feedback and the recommendation for publication of our manuscript.

CrSb is a representative altermagnet in the emerging class of collinear magnets. The manuscript by G. Yang and coworkers reports on the experimental and theoretical aspects of the CrSb altermagnet. This is an interesting follow-up study of the pioneering work by S. Reimers et al. (Ref.41) because:

- (i) the ARPES measurements were carried out at different locations of the CrSb Brillouin zone
- (ii) the authors compare their experimental results within the context of tight-binding model, thereby discussing the CrSb altermagnetic splitting by estimating the role of the mediation of the nearest neighbor Sb ions

Overall, the manuscript is a valuable contribution to characterizing the CrSb altermagnetism. However, this reviewer is not convinced that the manuscript is suitable for publication in Nature Communications. Several issues preclude publication in the current form. Here is a summary of the main issues I'd like to put forward for the general audience interested in CrSb altermagnetism:

- A) In the field of altermagnetism, there is an ongoing discussion regarding the role of the spin-orbit coupling (SOC). For example, in Ref.12 the experimental work concentrated on characterizing the electronic structure outside the nodal planes depicted in Fig.1, where the altermagnetic spin-splitting can be ignored. On one hand, the authors of this manuscript claim that SOC is weak for Cr 3d electrons and plays a minor role in the low-energy band structure. On the other hand, the theoretical calculations in Fig. 2g shows that for the "AM with SOC" case the spin polarization appears completely washed out compared to Fig.2k, which is the "AM without SOC". The fact that SOC is not needed does not mean that it is not present. What is the message here: does the SOC destroy the spin polarization? In addition, the discussion related to Fig. 2j "(NM) with SOC" is missing in the main text. It is unclear how is the non-magnetic (NM) case implemented and how does this connect to panels j and k in Fig.2?
- B) Unfortunately, the most important Ref.41 is only marginally mentioned in the manuscript. The authors in Ref.41 (presumably) intentionally concentrated on the P-Q-P rather than M- Γ -M directions in order to provide the experimental evidence for AM splitting. Concerning the point A, did the authors consider to calculate the "AM with SOC" to check the spin degeneracy predicted in their TB-model along the P-Q-P directions? Such a comparison would be extremely useful to elucidate the role of the AM splitting in CrSb 3D electronic band structure.
- C) Within the TB-approximation, loosely speaking, the atoms are free, whereas the electrons are tightly bounded. The fact that the AM nearest-neighbor effects in spin-splitting arises from strong third-nearest-neighbor hopping mediated by Sb ions is an interesting aspect of the manuscript. This message, however, comes out under significant computational machinery: starting from DFT (VASP), across PBE parametrization of GGA, to finally reach out to a TB-model Hamiltonian with maximally projected Wannier function method. The authors claim that their results "*show that altermagnetism plays a dominant role in the observed band structure, while the SOC effect is not very important*". The main problem I have here is that based on my critique in point A, the 1 eV spin-splitting in the M- Γ -M direction in Fig. 2g, is simply zero or close to zero. How is the 3NN AM splitting connected with the vanishing spin splitting indicated by green arrow in Fig.2g?

Furthermore, there are following minor points related with the manuscript:

- a) The ARPES data presented are in the UV-range, hence they are surface sensitive. Despite this, the 3D band maps at selected photon energies in Fig.2a-f faithfully reflect the bulk-sensitive DFT calculations. However, this is not the case for the AM-splitting discussed in Fig.2l-m: it is impossible to see the spin splitting in the indicated k_z -cut. Data summarized in Fig.3, however, do support this splitting. I would suggest mentioning this in text. As these data are related to CrSb g -wave altermagnetism, which is very difficult to observe in Fig.3a, I also suggest the authors to refer to recent work in <https://arxiv.org/abs/2405.12575>, where the 6-fold symmetry in the constant energy map is indeed well observed.
- b) In Fig.2, panels a,d,g,j,k I would suggest not using the Γ -bar M-bar notation, just like in panels b,e,h; because these ARPES data and calculations are at specific k_z values in the 3D band electronic band structure.
- c) In Fig.1d, I do not know how to interpret the $k_z = \pi c^*/2$ label

In conclusion, although the presented manuscript is a valuable contribution to CrSb altermagnetism, I am not convinced that it is suitable for a broader audience and thus for publication in Nature Communications in the present form. With adequate response to my critique A,B,C, however, I'd like to reconsider my recommendation.

I am pleased to say that the authors reacted to my critique, answering most of the questions I raised during the first review. With the additional SARPES data, the authors provided valuable information in understanding the antiferromagnetic (AM) spin splitting in CrSb. Based on the comments and critiques reacted/answered also to other reviewers, my opinion is that the manuscript is suitable for publication in Nature Communications. However, I take the liberty to address a few important revisions before acceptance.

Let me address the manuscript from a broader perspective first. The manuscript is, to a large extent, difficult to read. Fair enough, it combines spin- and angle-resolved photoemission (SARPES) mapping in 3D momentum space, which for AM systems is the experimental method of choice, with TB ab-initio calculations. As I mentioned during the first review round, these calculations give insight into specific hopping terms that show the relevance of 3NN terms. However, this part of the manuscript appears disconnected from the SARPES message. The problem is that to underpin this message in text with different MnTe and VSb₃S₆ systems, the outcome becomes rather speculative. For example, contrary to CrSb, in MnTe the AM spin-splitting at Fermi level (EF) is based on Te p-orbital states because the Mn d-states are deep in the valence band. VSb₃S₆ is a vdW 2D-like system, hence again significantly different in terms of orbital ordering. The systems indeed share the g-wave AM order parameter outside the nodal planes, but the role of the orbital interplay in the AM spin-splitting magnitude close to EF would need detailed TB-analysis of these systems, which is not the scope of the manuscript.

On the other hand, the unusual NN, NNN, and 3NN hopping amplitudes in CrSb stem from a dichotomy in “localized” and “delocalized” magnetism in AM systems, because the 3NN hopping appears to be mediated by Sb orbital states. Let me emphasize that, in contrast to semiconducting MnTe, CrSb is a metal, hence one would a priori expect an «itinerant» type of magnetism mediated with conduction band. In that sense, the TB-analysis indicates that CrSb becomes an «itinerant» AM in a very specific way. Did the authors consider the $q_2 + q_3$ terms arising from the direct and assistant hopping processes to be related to such a dual AM character?

Next, I address the SARPES part of the manuscript. The 3D k_z mapping summarized in Fig.2 is convincing, the panel m now reflects the peculiar k_z -dispersion quite well. The authors refer to AM spin-splitting for $k_z=0.28c^*$, where the “strong” AM splitting is expected at 80 eV photon energy. The relevant ARPES band maps are in Fig.2a and Fig.4a, which should be equivalent. However, it is evident that the data in Fig.2a is of better quality compared to Fig.4a. I understand this is because data in Fig.4a were measured from a different sample prepared for the SARPES experiment. If so, this should be mentioned in the text because the authors qualitatively managed to reproduce comparable ARPES band maps to further examine the CrSb AM spin-splitting.

Next, the arrows in the second derivative map in Fig.4b indicate the CrSb AM double-peak structure. If the “double-peak” structure refers to the red/blue arrows; and not the broad spectral intensity for $k_{//}<0$ and $k_{//}>0$, respectively, then the statement:

It is clear that both spin-up and spin-down MDCs exhibit a double-peak structure with large broadening, corresponding to the spin-up and spin-down bands labelled in Fig. 4b.

...is incorrect, because the broad antisymmetric MDC spectral weight does not resolve the double-peaks. Instead, the S_z up/down signal in Fig.4c features asymmetric spectral distribution for $k_{//}<0$ and $k_{//}>0$, respectively. The two bands for $k_{//}<0$ and $k_{//}>0$ are not resolved as in Fig.2a, and neither in the SARPES data in Fig.4d. Importantly, data in Fig.4c,d are missing the error margins. They would confirm that the double-peak structure can hardly be resolved in SARPES. Yet, the MDC has the expected spin-polarization asymmetry for $k_z=0.28c^*$ for $k_{//}<0$ and $k_{//}>0$. This is important because, within the MITM mirror plane, this MDC in AM spin-splitting must change sign. The ultimate proof of the CrSb AM spin-splitting would be to measure the same MDC for opposite $k_z=-0.28c^*$, which must have opposite S_z in MDC distribution. Did the authors manage to measure this dataset with different photon energy?

Minor issues:

- I would also recommend adjusting the cartoon figure in Fig.1d: the red and blue shades intuitively connect with the S_z spin polarization in the nodal planes discussed in the text, suggesting a symmetric spin polarization outside the $k_z=0$ nodal plane. This is misleading. The symmetric AM spin splitting can occur only in the diagonal direction $-\Gamma L$. If the authors did not intend to label the AM spin polarization of the four nodal planes in the cartoon figure, I would recommend using different color shades.
- *The two opposite-spin sublattices in altermagnets should be connected by a (screw) n -fold rotation (C_n): n cannot be an arbitrary integer, for CrSb it is 6*
- *When SOC comes into play, the $SU(2)$ symmetry is broken and the bands with opposite spins will couple to each other.* I do not understand the meaning of the $SU(2)$ symmetry, it is not mentioned nor explained in the text.
- *Indeed, if the SOC in CrSb were strong, it would transform the aforementioned nodal planes into a nodal line along the $\Gamma - A$ direction [40].* CrSb possesses topological Weyl nodal structures which do not necessarily require strong SOC according to Ref.68. Therefore, I recommend dropping this statement.

In conclusion, the manuscript contains valuable findings in understanding the CrSb altermagnetism. I recommend the publication for Nature Communications in an improved version of the manuscript along the lines mentioned above. Also, the manuscript is difficult to read for a broad audience. In that sense, I would recommend revising the paragraph on the «Mechanism behind the large altermagnetic splitting». It is advisable to reconsider the interpretation of the TB model of CrSb because the way this message extrapolates to MnTe and VSb3S6 is speculative. If the authors have more SARPES data supporting the ultimate AM splitting in 3D CrSb mentioned above, I would recommend using these data in this manuscript.